# Benefits of Max Pooling in Neural Networks: Theoretical and Experimental Evidence

**Kyle Matoba** *kyle.matoba@epfl.ch*
*Machine Learning Group, Idiap*
*Computer Science Department, École polytechnique fédérale de Lausanne*

**Nikolaos Dimitriadis** *nikolaos.dimitriadis@epfl.ch*
*Computer Science Department, École polytechnique fédérale de Lausanne*

**François Fleuret** *francois.fleuret@unige.ch*
*Computer Science Department, University of Geneva*
*Machine Learning Group, Idiap*
*Computer Science Department, École polytechnique fédérale de Lausanne*

**Reviewed on OpenReview:** *https://openreview.net/forum?id=YgeXqrH7gA*

## Abstract

When deep neural networks became state of the art image classifiers, numerous max pooling operations were an important component of the architecture. However, modern computer vision networks typically have few, if any, max pooling operations. To understand whether this trend is justified, we develop a mathematical framework analyzing ReLU based approximations of max pooling, and prove a sense in which max pooling cannot be replicated. We formulate and analyze a novel class of optimal approximations, and find that the residual can be made exponentially small in the kernel size, but only with an exponentially wide approximation.

This work gives a theoretical basis for understanding the reduced use of max pooling in newer architectures. It also enables us to establish an empirical observation about natural images: since max pooling does not seem necessary, the inputs on which max pooling is distinct – those with a large difference between the max and other values – are not prevalent.

## 1 Introduction

When convolutional neural networks first became state of the art image classifiers in the early 2010s, max pooling featured prominently. For example, in VGG (Simonyan & Zisserman (2015)) and AlexNet (Krizhevsky et al. (2017)). Largely coincident with this period, however, Springenberg et al. (2015) argued that max pooling operations were not necessary because strided convolutions composed with nonlinearity is simpler and more flexible. And subsequent architectures in such a direction – ResNets (He et al. (2016)) have a single max pooling layer, and some – such as InceptionV3 (Szegedy et al. (2016)) and mobilenetV3 (Howard et al. (2019)) – have none at all.[1] Max pooling is mostly omitted in the attention-based image classifiers that began to be used in the early 2020s, such as the Vision Transformer (Dosovitskiy et al. (2021)). Examining whether max pooling can truly be dropped from the toolbox of neural network image classifiers is worthwhile because it would ease architecture design and reduce development burden.[2]

This paper examines whether max pooling is a remnant of an earlier era when image classifiers were motivated by the visual cortex. We find for some inputs, max pooling constructs very different features than an optimal

---

[1] All statements about historical models refer to their implementation in Torchvision (Marcel & Rodriguez (2010)), described at https://pytorch.org/vision/stable/models.html.

[2] For example, in the modern era of extremely specialized hardware, it is common for primitive operations to be reimplemented several times for different computational "backends".

approximation by ReLUs, thus it expresses a different inductive bias and can sometimes be appropriate. We derive comprehensive bounds on the error realized by approximating max functions with the composition of ReLU and linear operations, and find that a simple divide and conquer algorithm cannot be improved upon. Hence, accurate approximations must be computationally complex.

Our result does not imply that omitting max pooling from image classifiers is wrong. Rather, it says when it *could be wrong*. Max pooling will be strictly more expressive than a ReLU-based approximation on inputs with a large difference between the maximum and other values within pools.

Our contributions are (1) introducing a novel generalization of max pooling (Section 4), (2) proving that the max function cannot be represented by simpler elements of this function class (Theorem 4), (3) analyzing experimentally the size and quality of approximations (Section 5), (4) formulating a notion of separation for max pooling (Theorem 1, Theorem 2), and (5) connecting mathematically the average of subpool maximums with order statistics (Theorem 3).

## 2 Background and Related work

Early work examined the neurological antecedents of (artificial) neural networks, and experiments with mammal's eyes and brains have shown max pooling to be a biologically plausible operation (Fukushima (1980); Riesenhuber & Poggio (1999)). And practice has followed this observation, with max pooling being important when deep neural networks first began achieving competitive performance. For example, AlexNet (Krizhevsky et al. (2017) dominated the ImageNet Large Scale Visual Recognition Challenge (ILSVRC) 2012 (Russakovsky et al. (2015)), and had max pooling in more than half of its "blocks".[3] However, the ILSVRC 2017 leaderboard was dominated by Resnets, for which max pooling is not central. Similarly with DawnBench (Coleman et al. (2018)), where a more transparent source code distribution model makes it simple to verify that many models feature only a single max pooling layer. This move away from max pooling was consistent with an influential study arguing that strided convolution composed with nonlinearity is preferable to max pooling because it can fit a max pooling mapping if appropriate, and can seamlessly fit another functional form if not (Springenberg et al. (2015)).

We examine this assertion by questioning whether (1) max pooling can be simplified to ReLU nonlinearity, and if not, (2) what pattern in the data would make max pooling and ReLU comparably expressive *empirically*.

Our work is preceded by a literature examining the expressiveness of piecewise linear networks in terms of bounds on the number of linear regions they express, for example Montúfar et al. (2014), Serra et al. (2018), Hanin & Rolnick (2019), and Montúfar et al. (2022). Several of these studies examine the maxout activation function (Goodfellow et al. (2013)), which is essentially equivalent to our mathematical model of max pooling. Maxout networks generally express many more linear regions than equally-sized networks using ReLU activations.

Compared to these studies, we assess the increased expressivity of max pooling (or maxout) networks as the error inherent to approximating a single max pooling operation with ReLUs. This makes our analysis more direct and practically interpretable. Though, because it may be that the outputs of a network using max pooling is efficiently approximated, even though an individual neuron could not, our conclusions may also need a bit more thought when transposed to architecture design. Our more narrow assumptions in turn suggest a potentially fruitful direction of inquiry into maxout activations: eschew general maxout assumptions for more restrictive architectures consistent with max-pooling (e.g. that the output is stricly smaller than the input, inputs are grouped into disjont "pools", etc.). These stronger assumptions may be the missing link since maxout, although more general, never gained much traction (it is not implemented natively in PyTorch, for example, and practically never seen in the wild).

Like most related work, our work easily extends to many other (elementwise) piecewise linear activations such as hard $\tanh = x \mapsto \mathrm{ReLU}(x+1) - 1 - \mathrm{ReLU}(x-1)$ ((Collobert, 2004)), leaky $\mathrm{ReLU} = x \mapsto \mathrm{ReLU}(x) -$

---

[3]The leaderboard is here: `https://image-net.org/challenges/LSVRC/2012/results.html`, AlexNet is a product of the "SuperVision" team.

| Depth | Function Class | Result |
|---|---|---|
| $\lceil \log_2 d \rceil$ | small width | exact trivial Theorem 2 |
| 1 | width $\uparrow \infty$ | approximation possible (Hornik, 1991; Cybenko, 1989) |
| $\lceil \log_2 d \rceil - 1$ | width $\uparrow \infty$ | exact impossible on $\mathbb{R}^d$ (Hertrich et al., 2021) |
| $\lceil \log_2 d \rceil - 1$ | $\mathcal{M}_d(R)$ | exact impossible on $[0,1]^d$ (Theorem 4) |
| $\lceil \log_2 d \rceil - 1$ | width $\uparrow \infty$ | approximation experimentally difficult on $[0,1]^d$ (Section 5) |

Table 1: A taxonomy of approximations to the max function by linear-ReLUs blocks in $d$ dimensions. $\mathcal{M}_d(R)$ is the function class introduced in Section 4. The quality of the approximation depends crucially on depth, and as far as we are aware, this is the first paper to examine the finite width case.

$.01 \times \text{ReLU}(-x)$ ((Maas et al., 2013)), $\text{ReLU6} = x \mapsto \text{ReLU}(x) - \text{ReLU}(x - 6)$ ((Krizhevsky, 2012)), and hard sigmoid $= x \mapsto (\text{ReLU}(x + 3) - \text{ReLU}(x - 3))/6$ ((Courbariaux et al., 2015)) and does not apply to many other sources of nonlinearity, such as tanh. However, there are standard techniques for extending work on piecewise linear activations to general activations based on bounding the difference to a piecewise linear approximation, such as Zhang et al. (2018).

Attention mappings do not fit within the scope of our analysis, being neither piecewise linear, nor simply approximated by a piecewise linear mapping. Nonetheless, we see that the recent move towards of fewer max pooling layers has been continued in the emergence of transformer-based image classifiers. The Vision Transformer (Dosovitskiy et al. (2021)), for example, uses no max pooling layers.

Hertrich et al. (2021) is closely related to the theoretical component of our work. This study addresses aspects of ReLU-based approximations to the max function, and proves a technical sense in which max pooling is more expressive than ReLUs. However, their proof technique relies crucially upon the discontinuity at zero, a dynamic not shared by our analysis, which is restricted to nonnegative inputs. We give precise error bounds, and are able to compare the errors of a large parameterized family of approximations with considerable precision to offer a more precise grasp of the practical tradeoff that applied approximations entail. Table 1 summarizes several results on the approximation of the maximum function by linear-ReLUs blocks.

Boureau et al. (2010) is an early work comparing average and max pooling from an average-case statistical perspective. They also identify the input dimensionality as a key factor in complexity. In their framework, sparse or low-probability features correspond to the corners of the input domain in our analysis.

Grüning & Barth (2022) find that *min* pooling can also be a useful pooling method, a finding that supports and is rationalized by our analysis in terms of the quantiles of the input to the pooling method. Essentially, if the true data is strongly determined by the nonlinear behavior of quantiles, then ReLU-based approaches are relatively disadvantaged.

## 3   The complexity of max pooling operations

In this section, we prove that in a simplified model, max pooling *requires* depth – multiple layers of ReLU nonlinearity are necessary to effect the same computation, and more layers are needed for larger kernel sizes.

### 3.1   Simplifications

In the design of deep learning architectures, max pooling reduces dimensionality by summarizing the values of spatially nearby inputs. To simplify, we examine the approximation of a max *function*, putting aside "pooling"-specific considerations like stride, padding, and dilation which that are ultimately linear pre-and post-processing. We summarize the size of the input using the term "order" so that, for example, the maximum over a $3 \times 3$ input is an order 9 max function. In this sense, our abstract model of max pooling is akin to the *maxout* activation from Goodfellow et al. (2013). We treat as a general linear function any operator that is linear at inference time, such as batch normalization, convolution, reindexing, or average pooling. Finally, we discuss only ReLU nonlinearities.

Such DNNs, say $f$, can be written as

$$f = (x \mapsto W_k x + b_k) \circ \text{ReLU} \circ \ldots \text{ReLU} \circ (x \mapsto W_0 x + b_0) \tag{1}$$

where $W_j, b_j, j = 0, 1, \ldots, k$ are the weights and biases of the linear layers, and $k$ is the depth. For brevity, we call these functions feedforward networks and leave all the other qualifiers implicit. Feedforward networks are convenient and pedagogically tidy because they concisely reduce all aspects of the architecture to just the number of layers and their widths.

## 3.2 Max pooling as a feature-builder

Telgarsky (2016) showed that deep neural networks cannot be concisely simulated by shallow networks. Their approach is to demonstrate a classification problem that is easy for deep networks, but is provably difficult for shallow networks. We do similarly by building a test problem on which max pooling succeeds and ReLU fails. First, we investigate the appropriate metric for comparison. Theorem 1 shows that for any dimensionality, a narrow feedforward network with a single source of nonlinearity can emit the same output as max pooling. Thus, prediction accuracy is not the correct metric by which to compare nonlinearities.

**Theorem 1.** *There exists a feedforward network $f : \mathbb{R}^d \to \mathbb{R}$ with $d$ hidden neurons such that for all $\xi \in \mathbb{R}$, $f(x - \xi) \leq 0 \iff \max\{x_1, \ldots, x_d\} \leq \xi$.*

*Proof.*

$$\max\{x_1, \ldots, x_d\} \leq \xi \iff x_1 \leq \xi \text{ and } \ldots \text{ and } x_d \leq \xi \iff \sum_{k=1}^{d} \text{ReLU}(x_k - \xi) \leq 0.$$

$\square$

Max pooling is used in the construction of intermediate layer features, and not directly in the computation of final logits. Thus, we examine the ability of a feedforward ReLU network to achieve the same real-valued output as a max pooling operation with $L_\infty$ error. And since diminishing the representation capacity of a single neuron necessarily reduces the expressivity of an entire network, we focus on the expressivity of a single neuron.

## 3.3 Computing max using ReLU

Theorem 1 gave a positive result on the complexity of functions that composition of linear and ReLU functions can represent. In this section we begin developing our negative results.

The maximum of two values can be computed using the relationship

$$\max(a, b) = (\text{ReLU}(a - b) + \text{ReLU}(b - a) + a + b)/2. \tag{2}$$

It seems that there is no tractable generalization of this formula for dimension $d > 2$. For example, in Appendix A, we give the analogous equation for the ternary case, and see that it is not linear in ReLU features. Building upon this, the appendix contains a heuristic argument that expressing the maximum of more than four variables without function composition may not be possible in general.

Nevertheless, Theorem 2 shows how with additional depth a feedfoward network can compute the maximum of many variables by recursively forming pairwise maxes.

**Theorem 2.** $\max: \mathbb{R}^d \to \mathbb{R}$ *can be written as a $\lceil \log_2(d) \rceil$-hidden layer feedforward network, where the $k$th hidden layer has width $2^{2 + \lceil \log_2(d) \rceil - k}$.*

*Sketch of Proof.* A variant of Equation 2 that is applicable to feedforward networks is

$$\max(x, y) = (g \circ \mathrm{ReLU} \circ f)(x, y) \text{ where}$$

$$f(x, y) = \begin{pmatrix} +x - y \\ -x + y \\ +x + y \\ -x - y \end{pmatrix} \tag{3}$$

$$g(x) = (x_{0:n/4} + x_{n/4:n/2} + x_{n/2:3n/4} - x_{3n/4:n})/2.$$

Here $x_{n_1:n_2}$ means the $n_1$th through the $(n_2 - 1)$th elements of $x$ (inclusive), and $n$ is the dimension of the input. $f$ and $g$ are linear. At the cost of quadrupling every layer width, ReLU can evaluate pairwise maxes and $\lceil \log_2(d) \rceil$ iterations of pairwise maxima can compute the maximum of $d$ variables. $\qquad \square$

As an example, the max of five variables is simply written as three iterations of pairwise maxes:

$$\max(x_1, x_2, x_3, x_4, x_5) = \max(z_1, z_2)$$
$$\text{where } z_1 = \max(z_3, z_4), z_2 = \max(z_5, z_6)$$
$$\text{where } z_3 = \max(x_1, x_2), z_4 = \max(x_2, x_3), z_5 = \max(x_3, x_4), z_6 = \max(x_4, x_5).$$

Theorem 2 is an upper bound on the width and depth necessary to evaluate a max function. Corresponding lower bounds are more intricate. Table 1 outlines various possible converses, and the next section discusses the function class that characterizes our innovation.

## 4 The class of subpool max averages, $\mathcal{M}_d(R)$

Our work pertains to a particular function class, $\mathcal{M}_d(R)$. In Section 4.1 we describe $\mathcal{M}_d(R)$, and in Section 5 we justify the relevance of this function class to deep learning.

### 4.1 Subpool maxes

For a vector $x \in \mathbb{R}^d$ and index set $J \subseteq \{1, \ldots, d\}$ we term $\max\{x_j : j \in J\}$ the $J$-subpool max of $x$. For example, if $x = (3, 2, 10, 5)$, then the $\{1, 2, 4\}$-subpool max of $x$ is $\max(3, 2, 5) = 5$. $J$-subpool maxes are generalizations of the max function that trade off complexity and accuracy in the sense that for $J_1 \subseteq J_2$, the $J_1$-subset max is simpler to compute than the $J_2$-subset max, but the error of a $J_1$-subpool max will always be at least that of a $J_2$-subpool max.

Let $C(j, r, d)$ denote the $j$th (out of $\binom{d}{r}$) subset of $\{1, \ldots, d\}$ of size $r$ in the lexicographic ordering. For example, $C(1, 2, 3) = \{1, 2\}, C(2, 2, 3) = \{1, 3\}$ and $C(3, 2, 3) = \{2, 3\}$. To keep the notation manageable, let $\omega_{jr}(x)$ denote the $C(j, r, d)$-subpool max of $x$ (the dimension $d$ can be inferred from the size of $x$). And in order to more elegantly model a constant intercept, let the $C(1, 0, d)$-subpool max be $= 1$.

Linear combinations of $J$-subpool maxes are a natural class of estimators to the max function. We organize subsets $J$ by (1) the size of the subset, $r$, and (2) the subset index of that size, $j$. For $R \subseteq \{0, 1, \ldots, d-1, d\}$ let

$$\mathcal{M}_d(R) = \left\{ x \mapsto \sum_{r \in R} \sum_{j=1}^{\binom{d}{r}} \beta_r^j \omega_{jr}(x) : \beta_r^j \in \mathbb{R} \right\} \tag{4}$$

be the set of all linear combinations of $r$-subpool maxes, $r \in R$. Let a general element of $\mathcal{M}_d(R)$ be called an $R$-estimator. Theorem 4 shows that $\max \notin \mathcal{M}_d(\{0, 1, \ldots, d-1\})$, with a bound on the $L_\infty$ error from this function class.

First, however, we present Theorem 3.

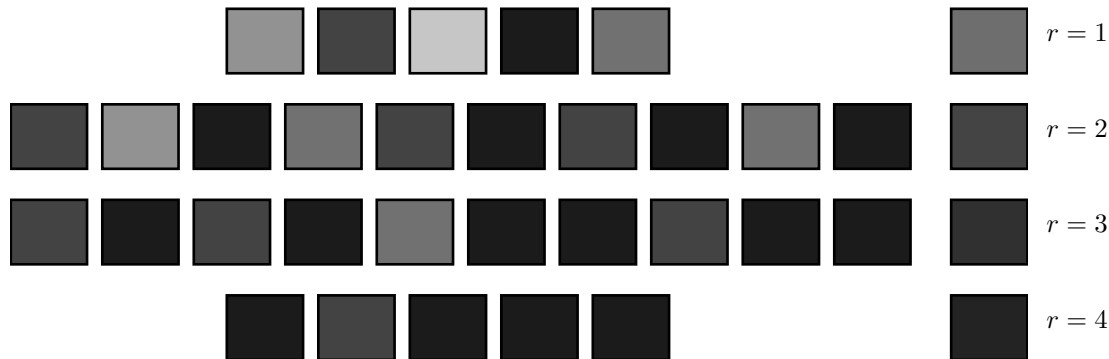

Figure 1: Averaged subpool maxes for $d = 5$. The numerical value of each node is presented as inverted grayscale. Each row $r = 1, 2, 3, 4$ presents on the left the averaged subpool maxes over pools of order $r$. As the order of the subpool maxes grows, the average value (on the right) grows darker towards the actual max.

**Theorem 3.** *Let $S(x; r, d) \triangleq \frac{1}{\binom{d}{r}} \sum_{j=1}^{\binom{d}{r}} \omega_{jr}(x)$ be the average of all subpool maxes of $x \in \mathbb{R}^d$ of order $r$. Let $x_{(j)}$ (the subscripts being enclosed in parentheses) denote the $j$th largest element of a vector $x$ (order statistics notation).*

$$S(x; r, d) = \frac{1}{\binom{d}{r}} \sum_{j=1}^{d-r+1} \binom{d-j}{r-1} x_{(j)}. \tag{5}$$

*Sketch of Proof.* $x_{(j)}$ is the largest value within a subpool if and only if all indices less than $j$ are excluded from that subpool and $j$ is not excluded. For a subpool of size $r$ the $r - 1$ remaining values must be amongst the $d - j$ values $x_{(j+1)}, \ldots, x_{(d)}$. Thus, amongst all subpools of size $r$, there will be $\binom{d-j}{r-1}$ in which the largest value is $x_{(j)}$. $\square$

The average of subpool maxes of an order $r \in R$ give a summary of the quantiles of the distribution via a particular weighted average, with better fidelity to the max for larger $r$. For example $S(x; 1, d)$ is a simple average, but $S(x; d-1, d) = ((d-1)x_{(1)} + x_{(2)})/d$, is mostly the max value, with only the second-largest value contributing. The idea is demonstrated further in Figure 1, with the individual subpool maxes easily tied back to the inputs via shading, and the higher-order subpool max averages increasingly resembling the actual max. As another demonstration, at the point $x = \begin{pmatrix} 0 & 0 & \ldots & 1 \end{pmatrix}$, the error of an order-1 approximation is $x_{(1)} - S(x; 1, d) = 1 - 1/d$. While for an order $d-1$ approximation it is much less, at $x_{(1)} - S(x; d-1, d) = 1/d$.

## 4.2 Optimal approximation over $\mathcal{M}_d(R)$

Now we present our main result, Theorem 4, gives the minimal errors achievable with approximations in $\mathcal{M}_d(R)$, for various $R$ over the $d$-dimensional unit cube. For example, Equation 9 states that the best $L_\infty$ error achievable using only terms from $R = \{0, 1, d-1\}$ (a constant term, the mean, and $S(x; d-1, d)$) is $1/(2(d+1))$ using $\begin{pmatrix} \beta_0 & \beta_1 & \beta_{d-1} \end{pmatrix} = \begin{pmatrix} 1/2 & -d/(d-2) & d(d-1)/(d-2) \end{pmatrix} /(d+1)$.

**Theorem 4.** *Let $||f||_\infty$ denote the the $L_\infty$ norm over the unit cube: $||f||_\infty = \sup_{x \in [0,1]^d} |f(x)|$. Let $\mathrm{dist}(R) = \min_{m \in \mathcal{M}_d(R)} ||m - \max||_\infty$, where $\mathcal{M}_d(R)$ is as defined in Equation 4.*

$$\text{dist}(\{0,1\}) = \frac{1}{2}\frac{d-1}{d} \tag{6}$$

$$\text{dist}(\{d-1\}) = 1/(2d-1) \tag{7}$$

$$\text{dist}(\{0, d-1\}) = 1/(2d) \tag{8}$$

$$\text{dist}(\{0, 1, d-1\}) = 1/(2(d+1)) \tag{9}$$

$$\text{dist}(\{0, 1, d-2, d-1\}) = 1/d^2 \tag{10}$$

$$\text{dist}(\{0, 1, 2, \ldots, d-1\}) = 1/2^d. \tag{11}$$

*Sketch of Proof.* The idea of the proof is as follows. We first establish symmetry as a property of any optimal estimator.

Then we assume that the $L_\infty$ norm of the error is characterized by $|R|$ nonzero corners of the unit cube and zero. Under this conjecture, the norm is optimized by evaluating the error as a function of the coefficients, and choosing coefficients which equate them. Generally, changing coefficients will increase the error at one point and lower it at others, and balancing these effects characterizes optimality. For example with $R = \{0, d-1\}$, the optimal coefficients are $(\beta_0, \beta_{d-1}) = (1/(2d), 1)$. At $x = (0, \ldots, 0), (1, 0, \ldots, 0)$ and $(1, 1, \ldots, 1)$ the error of this estimator is clearly $1/(2d)$. Given the conjectured error and coefficients, we then prove it is optimal by contradiction: by showing that any other coefficients attain a higher criterion at some $x$.

To continue the example from the proof of Equation 8, the only way that both $-1/(2d) < 1 - \beta_0 - \beta_{d-1}$ (the condition at $x = (1, \ldots, 1)$) and $1 - \beta_0 - \beta_{d-1}(d-1)/d < 1/(2d)$ (the condition at $x = (1, 0, \ldots, 0)$) is for $\beta_0 > 1/(2d)$. But this clearly precludes the error from being $< 1/(2d)$ at $x = (0, 0, \ldots, 0)$.

$\square$

We see that even with $R = \{0, 1, 2, \ldots, d-1\}$, the largest $R$ that does not include $d$, the error is not zero, meaning essentially that insufficiently deep networks in $\mathcal{M}(R)$ cannot model max pooling, and giving a partial converse to Theorem 2.

Equation 6 gives a linear model as a baseline for the max. The error is high, and in higher dimensions, not much different to a constant model (obviously, the least error that an intercept alone can obtain is $.5 = \text{dist}(\{0\})$). Equation 7 shows that error declining with dimensionality is achievable with a higher order term. Contrasting Equation 8 to Equation 7 quantifies the additivity of the intercept. Including an intercept helps, but primarily in low dimensions, which makes sense as an intercept does not scale with dimensionality. Since the intercept requires negligible computation, we assume its inclusion subsequently. Including the grand mean as a feature entails no meaningful further nonlinearity, and reduces the error from $1/(2d)$ to $1/(2(d+1))$ (Equation 9), thus we assume its inclusion subsequently.

Equation 10 is important for understanding how the error falls with the addition of further strongly dimension-sensitive terms: appending $d-2$ to $\{0, 1, d-1\}$ improves the rate of convergence from $O(1/d)$ to $O(1/d^2)$. Equation 11 gives the best-case rate of convergence: if *all* lower order terms are included, then the error is $O(1/2^d)$. Contrasting this with Equation 10, we see that the inclusion of many lower order terms is apparently necessary for low error. Qualitatively, Equation 11 implies that $\max \notin \mathcal{M}_d(\{0, 1, \ldots, d-1\})$, though it can be approximated well within the function class.

Let $f_R^\star : \mathbb{R}^d \to \mathbb{R}$ denote the optimal estimate based on terms in $R$. $L_\infty$ error can be high, even if $f_R^\star \approx \max$ on most of the domain. If the the high error could not be realized in practice, because the measure of the domain on which it arises is miniscule, then our result would be only a technicality with little practical relevance. In the literature studying the number of linear regions in a piecewise linear network, this distinction is recognized as the difference between the maximum and average number of linear regions (Tseran & Montúfar (2021)). Theorem 5 shows that this is not the case, with a lower bound on the measure of a set on which the $L_1$ norm is high.

**Theorem 5.** *For $\epsilon < \text{dist}(R)/2$, let $W(\epsilon; R) = \{x \in [0,1]^d : |x_{(1)} - f_R^\star(x)| \geq \epsilon\}$ denote the subset of the unit cube where the error of $f_R^\star$ is at least $\epsilon$. Then for all $R$ with $0 \in R$, $\text{vol}(W(\epsilon; R)) \geq (\text{dist}(R)/2 - \epsilon)^d$, where vol is the Lebesgue measure over $[0,1]^d$.*

In Appendix C we solve for the $L_2$ error of Equation 11 and find that it is also not zero. Thus: not only can the error be high, it is moreover high on average, and at many points on the domain.

In fact, the entire distribution of $x_{(1)} - f_R^\star(x)$ can be derived, and we do this in Appendix D. For simplicity, we here phrase only the headline result: the error is almost surely nonzero.

**Theorem 6.** *If $x$ is uniformly distributed over the unit cube, then $\text{vol}(\{x \in [0,1]^d : |x_{(1)} - f_R^\star(x)| = 0\}) = 0$.*

## 5 Experimental evidence on the relevance of $\mathcal{M}_d(R)$

The theoretical analysis of this paper would be diminished if a class of feedforward networks more general than $\mathcal{M}_d(R)$ could achieve significantly less error. This section presents experimental evidence that $\mathcal{M}_d(R)$ is an adequate proxy for all networks of the same depth in approximating the max function. It does this by showing that additional capacity does not appear to improve the quality of the approximation.

Formally, for $f$ given in Equation 1, let $w(f) \in \mathbb{N}^k$ contain the number of columns of $W_1, \ldots, W_k$ – the widths of the hidden layers. Appendix F presents a concrete algorithm for representing $f_R^\star$ in the form of Equation 1. For example, via this procedure with $d = 9$ and $R = \{0, 1, 7, 8\}$, $f_R^\star$ can be computed by a feedforward network with hidden layer widths $w(f_R^\star) = (78, 122, 182)$. Finally, let $\mathcal{G}_{d,k}$ denote the set of all depth $k$ feedforward DNNs $\mathbb{R}^d \to \mathbb{R}$, and for $\mu > 0$ let

$$\mathcal{G}_d(R, \mu) = \{g \in \mathcal{G}_{d, \lceil \log_2(\max R) \rceil} : w(g) \leq \mu \times w(f_R^\star)\} \tag{12}$$

be the set of all neural networks that are at most $\mu$ times as wide as $f_R^\star$, the optimal estimator in $\mathcal{M}_d(R)$.

Since $\mathcal{M}_d(R) \subseteq \mathcal{G}_d(R, 1), \mu_1 < \mu_2 \implies \mathcal{G}_d(R, \mu_1) \subseteq \mathcal{G}_d(R, \mu_2)$ and $\lim_{\mu \uparrow \infty} \mathcal{G}_d(R, \mu) = \mathcal{G}_{d, \lceil \log_2(\max R) \rceil}$, $\mathcal{G}_d(R, \mu)$ represent a parameterized interpolation between $\mathcal{M}_d(R)$ and all networks of a given depth and $\mu$ is our notion of "capacity" when we speak of adding capacity to $\mathcal{M}_d(R)$.

Our rhetorical point will be supported if a DNN does not achieve low error. This is awkward because failing to achieve low error on a deep learning task is not difficult, for example it could result from a coding mistake or a bad hyperparameterization. That said, given the simplicity of the modelling problem, excluding bugs is reasonably simple – we code it in an idiomatic fashion, check that the code can solve problems that should be solvable (for example, replacing the max with the mean), and transparently release the source code. We endeavor to show that our results are not caused by poor modelling choices with extensive ablation studies in the appendix.

There is an important precedent for this type of result in machine learning experiments: whenever a paper shows that some method is bested by another, we must assume that the inferior method implemented correctly and reasonably. Thus, although demonstrating experimentally the correctness of a failing method requires a high standard of evidence, the nature of the argument is not inherently problematic.

### 5.1 Experimental setup

Independent test and train datasets are generated uniformly on the unit cube with 10,000 rows. $L_\infty$ loss is directly optimized by an Adam optimizer (with PyTorch default parameters). Results are similar with MSE loss. We use PyTorch default initialization. A batch size of 512 is used throughout, and the data set is shuffled over epochs. Training is run for a uniform 300 epochs. All computations are run on a mix of inexpensive, consumer-grade graphics processing units (GPUs), and all experiments described in this section can be run in a few GPU-days. We repeat all analyses over ten pseudorandom number generator seeds, with both the data being generated differently, and different randomness in the fitting (e.g. the shuffling over minibatches).

## 5.2 Results

Our experimental evidence consists of assessing how expressiveness of trained models $\in \mathcal{G}_d(R, \mu)$ depends on $\mu$. A DNN $\in \mathcal{G}_d(R, \mu)$ optimized to model $x \mapsto x_{(1)}$ is more expressive if it achieves a lower empirical test $L_\infty$ error. We term this quantity $\mathrm{ERR}(R, \mu)$. If expressiveness is not substantially increased ( $\iff \mathrm{ERR}(R, \mu)$ is not substantially decreased) for progressively greater $\mu$, then a lower bound on the error of approximating the max function over $\mathcal{M}_d(R)$ empirically also holds for $\mathcal{G}_d$. And this is what we see.

We focus on $\mu$ around 1, because this corresponds to the width of $f_R^\star$.[4] To prove our desired point that adding capacity does not significantly increase expressiveness we require only $\mu \geq 1$, however we include also results for $\mu < 1$ to help foster intuition. This is especially important as the literature on *neural tangent kernels* (NTK), for instance Allen-Zhu et al. (2019), shows that for a fixed dataset size, error falls from high levels for (not all the assumptions of the NTK, such as a very small learning rate, hold so the prescriptions of this model are indicative) small models, but rapidly levels off.

Neither the NTK nor universal approximation theorems (UATs) insure that that there is a network that achieves arbitrarily low error as $\mu \uparrow \infty$. However, UATs make make no guarantees about fitting.

We examine $R = \{0, 1, d - 1\}, \{0, 1, d - 2, d - 1\}$, and $\{0, 1, 2, \ldots, d - 2, d - 1\}$ as prototypical "small", "medium", and "large" approximations, respectively. We see that although larger models do empirically achieve lower errors, the overall flattening trajectory of their dependence on increased capacity is uniform.

For a single order $d$ max-pool layer the maximum expected number of regions is, from Tseran & Montúfar (2021), Theorem 8: $1 + (d - 1)d/2$. Contrast this with Montúfar et al. (2014)'s bound for a single-output, $d$-input fully connected network of the form Equation 1, of $(1 + w(f)_1) \times \ldots \times (1 + w(f)_k)$. Thus, by this metric, $\mu = 4$ more than suffices to model the any reasonable $d$.

Figure 2 presents our main experimental result: fitting error does not reliably fall as $\mu$ rises. Even quadrupling all layer widths does not noticeably increase expressiveness. Clearly the amount of data available to train models of differering capacities is an important determinant of performance. Figure 3 examines the effect of dataset size on training error for the $R = \{0, 1, 2, \ldots, d - 2, d - 1\}$ model. Here the $x$-axis is on a logarithmic scale in $\mu$ space. We see that although test error continues to reliably fall with the addition of more data (in line with both the NTK, which guarantees that the global minimum – here at most $1/2^d$ by Equation 11 should be achieved with enough parameters, data, and a low enough learning rate), there is a marked flattening for all dataset sizes – beyond a certain point, simply adding capacity does not make a more accurate model.

Figure 4 analyzes each model separately in greater detail. Shaded around the sample mean is the region encompassed by $\pm 1.96$ standard deviations. The max and min values over the ten seeds are also plotted in dotted linds. We observe that the min and max are roughly coincident with $\pm 1.96$ standard deviations above and below the sample mean, so the distribution of results over seeds is even more thin-tailed than a Gaussian. The train error is also plotted as a dashed, slightly lower than the test. The train error follows the same trajectory as test.

This analysis shows that greater width does not seem to decrease approximation error. From this, we conclude that although our main results are proven only for $\mathcal{M}_d(R)$, empirically they translate well to more general and powerful function classes within the space of all feedforward networks. Appendix G contains additional results in order to further augment and establish the robustness of this conclusion.

## 5.3 The complexity of $\mathcal{M}_d(R)$

Equation 7 and Equation 11 represent quite disparate orders of estimation error. This is because $\mathcal{M}_d(\{0, 1, 2, \ldots, d - 1\})$ is much larger than $\mathcal{M}_d(\{0, 1, d - 1\})$. The most useful way to understand this

---

[4]There are however non-width reasons that, even for $\mu = 1$, $\mathcal{M}_d(R) \subsetneq \mathcal{G}_d(R, 1)$. One is that $\mathcal{G}_d(R, \mu)$ allows intercepts in the linear layers, despite not being present in $f_R^\star$ (cf. Appendix F). $\mathcal{G}_d(R, \mu)$ also imposes no low-rank structure on the weight matrices, despite a straightforward representation of the theoretical weights as the outer product of matrices roughly one quarter as large (via the quadrupling of layer sizes implied by Equation 3). The larger is the class of models that does not substantially reduce error, the more conservative are our experimental results and the stronger is our conclusion.

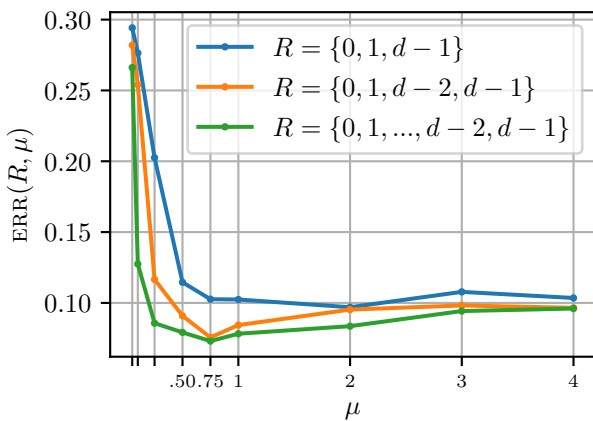 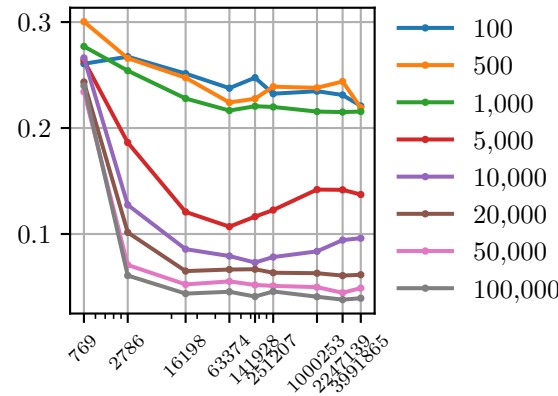

Figure 2: $\text{ERR}(R,\mu)$ ($y$-axis) does not continue to fall with greater $\mu$ ($x$-axis), meaning that expressivity is not increased with more capacity.

Figure 3: $x$-axis: number of parameters in a $R = \{0,1,\ldots,d-2,d-1\}$ model, $y$-axis: error. Each line is a dataset size, given in the legend on the right. Error declines with more data, but the pivotal pattern of Figure 2 is unchanged.

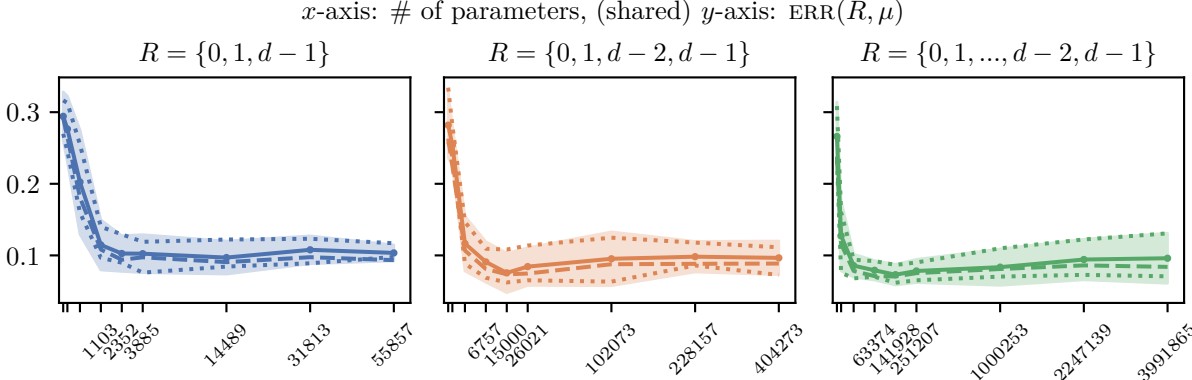

Figure 4: $x$-axes: number of parameters in $R$-estmators, $y$-axis: solid line: average test error, dashed line: average training error, dotted-lines: min / max over trials, shaded area: averaged test error $\pm 1.96$ standard deviations.

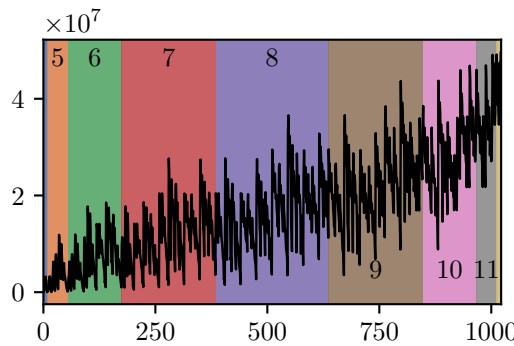

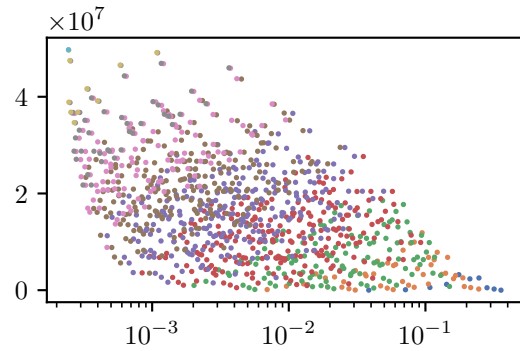

(a) Parameter count for $R$ ordered by size, then lexicographically within sizes. Different $|R|$ are denoted with differing background colors.

(b) (Logarithmic) $x$-axis: $L_\infty$ error, $y$-axis: parameter count. Marker color is as in Figure 5a.

Figure 5: Estimator size and error in $d = 12$.

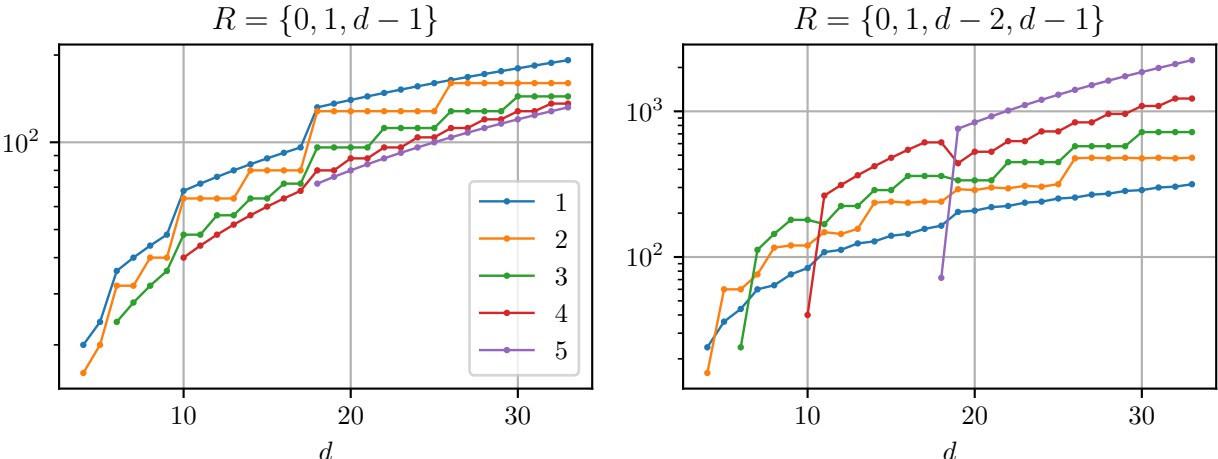

Figure 6: Model sizes for neural networks representing $f_R^\star$ for two different $R$. $y$-axis (logarithmic): hidden layer width, $x$-axis: $d$. Layer numbers by color are shown in the legend on the left plot. Depth of the requisite network at each dimension $d$ is implied by the number of lines present, for example a $d = 17, R = \{0, 1, d-1\}$ estimator has $w(f) = (96, 80, 72, 68)$.

is in terms of the network sizes that can model elements of $\mathcal{M}(R)$ for different $R$. Appendix F presents an algorithm for writing $f_R^\star$ as a feedforward network based on reusing lower order subpool maxes to evaluate higher order subpool maxes in a way that skips unneeded orders. This method is simple, and reasonably efficient, given the unavoidable necessity of computing essentially $O(\sum_{r \in R} \binom{d}{r})$ terms.

Figure 5 plots the number of parameters in a deep neural network representation of an $R$-estimator. Figure 5a relates the parameter counts to $R$. This ranges from less than 3500 parameters for $R = \{0, 1, 2\}$, to nearly 50 million for $R = \{0, 1, \ldots, d-1\}$. Figure 5b further relates the model size to the $L_\infty$ error bound computed in Appendix E to convey a sense of how many parameters are needed to achieve a given error. In both plots, we group models by $|R|$, indicated by color.

For another view on model size that offers more insightful on how network architecture is affected by dimension, Figure 6 plots the model sizes for networks implementing two $f_R^\star$ for two $R$, as a function of the model dimension, $d$. The depth of these models is always $\lfloor \log_2(\max(R) - 1) \rfloor + 1$ .

## 6 Conclusion

Motivated by a marked trend in the design of computer vision architectures, we have posed and answered the question: can max pooling be replaced by linear mappings composed with ReLU activations? And when would doing so give a model that is considerably different?

To do this, we developed a new notion of approximation and a new class of results that complements existing work on the number of linear regions in piecewise linear networks. We first established distance in intermediate feature space as the notion of comparison. Next, we established a simple baseline: max pooling with kernel size of $d$ can be computed by a block of $\log_2(d)$ narrow layers. We next introduced subpool max averages as a tractable class of approximators, and proved that the max function in $d$ dimensions cannot be written as the linear combination of subpool max averages of order $< d$, though the error can be made as low as $1/2^d$. By establishing experimentally that the class of subpool max averages was not significantly less expressive than more general function classes, we extended our analysis to wider networks not constrained to have a fixed weight pattern. As a byproduct of this analysis, we are able to also visualize the complexity of all approximators.

In future, we hope to further examine practical implications of this analysis, establishing experimentally that there me be some non-accuracy reasons to prefer max pooling, such as adversarial robustness. Some preliminary evidence on these conjectures is presented in Appendix H.1.

## Acknowledgements

We would like to thank Guillermo Ortiz-Jiménez, Apostolos Modas, Arnaud Pannatier, Suraj Srinivas, and the anonymous reviewers for their valuable feedback. KM is supported by the Swiss National Science Foundation under grant number FNS-188758 "CORTI". ND was supported by Swisscom (Switzerland) AG.

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

## A   In what sense is evaluating $\max(x_1, x_2, x_3, x_4, x_5)$ hard?

To motivate the differences of the max function with higher dimensionality, in this short section, we argue that there is unlikely to be a simple generalization of the correspondence between maximum and ReLU in $d = 2$ to general $d$. The idea is entirely due to Blatter (2011), though any mistakes in the interpretation are ours alone.

**Theorem 7.** *There is no algebraic expression for*

$$(x_1, x_2, x_3, x_4, x_5) \mapsto \max(x_1, x_2, x_3, x_4, x_5).$$

*Proof.* $\max(x_1, x_2, x_3, x_4, x_5)$ is a root of the polynomial $(x - x_1)(x - x_2)(x - x_3)(x - x_4)(x - x_5)$. Suppose that there was an algebraic expression for the largest of $(x_1, x_2, x_3, x_4, x_5)$, say $f(x_1, x_2, x_3, x_4, x_5)$ then, then the roots of $(x - x_1)(x - x_2)(x - x_3)(x - x_4)(x - x_5)/(x - f(x_1, x_2, x_3, x_4, x_5))$ could be found via the quartic equation, and we would have an algebraic expression for all five roots. However, Abel's Theorem states that there is no algebraic expression for general quintic polynomials.

$\square$

The argument is subtle, so consider the same argument applied to the maximum of two values. The two roots of $(x - x_1)(x - x_2)$ are well known to be $(x_1 + x_2)/2 \pm \sqrt{(x_1 + x_2)^2 - 4x_1x_2}/2$, and the larger corresponds to adding the discriminant:

$$\frac{(x_1 + x_2) + \sqrt{(x_1 + x_2)^2 - 4x_1x_2}}{2} = \frac{(x_1 + x_2) + \sqrt{(x_1 - x_2)^2}}{2} = \frac{(x_1 + x_2) + |x_1 - x_2|}{2}.$$

Which is a well-known trick for reasoning mathematically about the maximum of two variables. A corresponding expression comes from solving the cubic equation

$$\max(x_1, x_2, x_3) = \frac{1}{2} \frac{x_1(|x_1 - x_2| + |x_1 - x_3|) + x_2(|x_1 - x_2| + |x_2 - x_3|) + x_3(|x_2 - x_3| + |x_1 - x_3|)}{|x_1 - x_2| + |x_2 - x_3| + |x_1 - x_3|}$$
$$+ \frac{|x_1 - x_2| + |x_2 - x_3| + |x_1 - x_3|}{4}.$$

This equation is tractable because we can assess the min and max similar to above, and impute the third value from the average.[5] It is not clear what the analogous interpretation is for the quartic equation, though one presumably exists. However, for fifth and higher-order polynomials we cannot generally even write down an algebraic expression for the roots, so determining by inspection which will be the greatest seems unlikely.

## B   Proofs

### B.1   Proof of Theorem 4

*Proof.* Let $\mathrm{perm}(d)$ denote the set of all permutations of $\{1, 2, \ldots, d\}$, and let

$$\mathcal{M}_d^s(R) = \{m \in \mathcal{M}_d(R) : m(x_1, x_2, \ldots, x_d) = m(x_{\sigma_1}, x_{\sigma_2}, \ldots, x_{\sigma_d}) \text{ for all } \sigma \in \mathrm{perm}(d)\}$$

denote the restriction of $\mathcal{M}_d(R)$ to those elements that are invariant to a reordering of its arguments. Because max is symmetric in this sense, any optimal approximation to it must lie in $\mathcal{M}_d^s(R)$:

---

[5]See also the excellent exposition given at `https://math.stackexchange.com/a/89702/92999`.

**Theorem 8.** *For all R,*

$$\min_{m \in \mathcal{M}_d(R)} ||m - \max||_\infty = \min_{m \in \mathcal{M}_d^s(R)} ||m - \max||_\infty.$$

*Proof.* Assume otherwise, meaning that there is an $m$ that is not symmetric and an $x$ such that $m(x') < m^s(x)$ for all symmetric $m^s$ and all $x'$. In particular $m(x_\sigma) < m^s(x)$ for all permutations $x_\sigma$ of $x$. Thus

$$\frac{1}{d!} \sum_{\sigma \in \mathrm{perm}(x)} m(x'_\sigma) < m^s(x).$$

However, $x \mapsto \frac{1}{d!} \sum_{\sigma \in \mathrm{perm}(x)} m(x'_\sigma)$ is evidently symmetric, a contradiction.

$\square$

Thus, it is without loss of generality to optimize over $\mathcal{M}^s(R)$ rather than $\mathcal{M}(R)$, and we turn to operationalizing the symmetry assumption in terms of the coefficients. For $r > 1$, $\omega_{jr}(1_d - e_{dk}) = 1^6$ for all $k = 1, 2, \ldots, d$ and all $j = 1, \ldots, \binom{d}{r}$ since there is only a single non-one value. Thus all terms of order greater than 1 are equal, and the sums are equal by symmetry, so we necessarily have that for all $f \in \mathcal{M}_d^s(R)$, $\beta_1^1 = \beta_1^2 = \ldots = \beta_1^d$. Call this single value $\beta_1$, and (for $0, 1 \in R$)

$$\mathcal{M}_d^s(R) = \left\{ x \mapsto \beta_0 + \beta_1 S(x; 1, d) + \sum_{r \in R \setminus \{0,1\}} \sum_{j=1}^{\binom{d}{r}} \beta_r^j s(x; C(j, r, d)) : \beta_r^j, \beta_0, \beta_1 \in \mathbb{R} \right\}.$$

Repeating this process for pools consisting of entirely of 2, except for $3, 4, \ldots, d-1$ zeros in turn implies that the estimator must be a function of $S(x; r, d)$ alone, and not the individual terms of the sum separately:

$$\mathcal{M}_d^s(R) = \left\{ x \mapsto \sum_{r \in R} \beta_r S(x; r, d) : \beta_r \in \mathbb{R} \right\}. \tag{13}$$

So, the maximization over all $m \in \mathcal{M}_d(R)$ can be reduced to the maximization of $|R|$ scalar coefficients.

In the simplest terms, to prove Theorem 4, we want to show that

$$\min_{\beta_0, \beta_1} ||x_{(1)} - \beta_0 - \beta_1 S(x; 1, d)||_\infty = \frac{1}{2}\frac{d-1}{d} \tag{14}$$

$$\min_{\beta_{d-1}} ||x_{(1)} - \beta_{d-1} S(x; d-1, d)||_\infty = 1/(2d-1) \tag{15}$$

$$\min_{\beta_0, \beta_{d-1}} ||x_{(1)} - \beta_0 - \beta_{d-1} S(x; d-1, d)||_\infty = 1/(2d) \tag{16}$$

$$\min_{\beta_0, \beta_1, \beta_{d-1}} ||x_{(1)} - \beta_0 - \beta_1 S(x; 1, d) - \beta_{d-1} S(x; d-1, d)||_\infty = 1/(2(d+1)) \tag{17}$$

$$\min_{\beta_0, \beta_1, \beta_{d-2}, \beta_{d-1}} ||x_{(1)} - \beta_0 - \beta_1 S(x; 1, d) - \beta_{d-2} S(x; d-2, d) - \beta_{d-1} S(x; d-1, d)||_\infty = 1/d^2 \tag{18}$$

$$\min_{\beta_0, \beta_1, \ldots, \beta_{d-1}} ||x_{(1)} - \beta_0 - \beta_1 S(x; 1, d) - \ldots - \beta_{d-1} S(x; d-1, d)||_\infty = 1/2^d \tag{19}$$

where the $L_\infty$ norm is taken over values of $x \in [0, 1]^d$.

---

$^6\omega_{jr}(x)$ is defined in subsection 4.1, the $C(j, r, d)$-subpool max of $x$.

The proof of each is similar, to facilitate their analysis, we distill the essence of each into a template. There are three essential steps:

1. Conjecture coefficients, $\beta^\star$ with the help of Appendix E.

2. Lower bound: for any $f$ and for any set of points $P \subseteq [0,1]^d$,

$$\min_{\beta} \max_{x} |f(\beta, x)| \geq \min_{\beta} \max_{x \in P} |f(\beta, x)| = \max_{x \in P} |f(\beta^\star, x)|$$

where we prove the equality by contradiction: suppose there is a better $\beta'$, and derive a contradiction.

3. Upper bound: for any $f$,

$$\min_{\beta} \max_{x} |f(\beta, x)| \leq \max_{x} |f(\beta^\star, x)|.$$

Then we show that $\max_x |f(\beta^\star, x)| \leq \beta_0^\star$ by writing it as a linear combination of order statistics.

For Equation 14, let $\begin{pmatrix} \beta_0^\star & \beta_1^\star \end{pmatrix} = \begin{pmatrix} (d-1)/(2d) & 1 \end{pmatrix}$. Suppose that there were some $\begin{pmatrix} \beta_0' & \beta_1' \end{pmatrix}$ achieving a criterion $< \beta_0^\star$. Evaluating the error at $x = \begin{pmatrix} 1 & 1 & \dots & 1 \end{pmatrix}$ and $x = \begin{pmatrix} 1 & 0 & \dots & 0 \end{pmatrix}$, this implies

$$1 - \beta_0' - \beta_1'/d < +\beta_0^\star \text{ and } 1 - \beta_0' - \beta_1' > -\beta_0^\star \implies (d-1)(1 - \beta_0') < (d+1)\beta_0^\star$$
$$\iff \beta_0' > 1 - \frac{d+1}{d-1}\beta_0^\star = \beta_0^\star.$$

A contradiction to the condition at zero. Thus, $\min_{\beta_0, \beta_1} ||x_{(1)} - \beta_0 - \beta_1 S(x; 1, d)||_\infty \geq \beta_0^\star$.

On the other hand, from Theorem 3

$$x_{(1)} - \beta_0^\star - S(x; 1, d) = \left(1 - \frac{1}{d}\right) x_{(1)} - \frac{1}{d}\left(x_{(2)} + \dots + x_{(d)}\right) - \beta_0^\star \in [-\beta_0^\star, +\beta_0^\star].$$

$\square$

For Equation 15, let $\beta_{d-1}^\star = \frac{2d}{2d-1}$. Suppose that there were some $\beta_{d-1}'$ achieving a criterion strictly less than $1/(2d-1)$. Evaluating the criterion at $x = \begin{pmatrix} 1 & 1 & \dots & 1 \end{pmatrix}$ and $x = \begin{pmatrix} 1 & 1 & \dots & 1 & 0 \end{pmatrix}$, this is only possible if

$$1 - \beta_{d-1}' > -\frac{1}{2d-1} \text{ and } 1 - \beta_{d-1}'\frac{d-1}{d} < \frac{1}{2d-1} \iff \frac{2d}{2d-1} > \beta_{d-1}' \text{ and } \beta_{d-1}' > \frac{2d}{2d-1},$$

which is to say it is impossible. On the other hand, from Theorem 3

$$x_{(1)} - \frac{2d}{2d-1}S(x; d-1, d) = x_{(1)} - \frac{2d}{2d-1}\left(\frac{d-1}{d}x_{(1)} + \frac{1}{d}x_{(2)}\right)$$
$$= \frac{x_{(1)} - 2x_{(2)}}{2d-1} \in \left[-\frac{1}{2d-1}, +\frac{1}{2d-1}\right].$$

$\square$

For Equation 16, let $\begin{pmatrix} \beta_0^\star & \beta_{d-1}^\star \end{pmatrix} = \begin{pmatrix} 1/(2d) & 1 \end{pmatrix}$. Suppose that there are some $\begin{pmatrix} \beta_0' & \beta_{d-1}' \end{pmatrix}$ achieving a criterion $< \beta_0^\star$. Evaluating the error at $x = \begin{pmatrix} 1 & 1 & 1 & \dots & 1 \end{pmatrix}$ implies

$$-1/(2d) < 1 - \beta_0' - \beta_{d-1}' \iff \beta_{d-1}' < 1 - \beta_0' + 1/(2d). \tag{20}$$

Evaluating the error at $x = \begin{pmatrix} 1 & 0 & 0 & \ldots & 0 \end{pmatrix}$ implies

$$
\begin{aligned}
1 - \beta_0' - \beta_{d-1}'(d-1)/d < +1/(2d) &\iff 1 - \beta_0' - 1/(2d) < \beta_{d-1}'(d-1)/d \\
&\iff \frac{d}{d-1}\left(1 - \beta_0' - 1/(2d)\right) < \beta_{d-1}'.
\end{aligned} \tag{21}
$$

Combining Equation 20 and Equation 21

$$
\begin{aligned}
\frac{d}{d-1}\left(1 - \beta_0' - 1/(2d)\right) < 1 - \beta_0' + 1/(2d) &\iff \left(\frac{d}{d-1} - 1\right)(1 - \beta_0') < \frac{1}{2d}\left(1 + \frac{d}{d-1}\right) \\
&\iff (1 - \beta_0') < \frac{2d-1}{2d} \\
&\iff \beta_0' > \frac{1}{2d}.
\end{aligned} \tag{22}
$$

A contradiction to the condition at 0 that $|\beta_0'| < 1/(2d)$. On the other hand, from Theorem 3

$$x_{(1)} - \frac{1}{2d} - S(x; d-1, d) = \frac{1}{d}\left(x_{(1)} - x_{(2)}\right) - \frac{1}{2d} \in \left[-\frac{1}{2d}, +\frac{1}{2d}\right]. \tag{23}$$

$\square$

For Equation 17, let $\begin{pmatrix} \beta_0^\star & \beta_1^\star & \beta_{d-1}^\star \end{pmatrix} = \begin{pmatrix} \frac{1}{2} & \frac{-d}{d-2} & \frac{d(d-1)}{d-2} \end{pmatrix}\frac{1}{d+1}$. Suppose that there were a $\begin{pmatrix} \beta_0' & \beta_1' & \beta_{d-1}' \end{pmatrix}$ achieving a criterion less than $\beta_0^\star$, then the condition at $x = (1,1,\ldots,1), (1,1,0,\ldots,0), (1,0,\ldots 0)$ imply, respectively:

$$
\begin{aligned}
1 - \beta_0' - \beta_1' - \beta_{d-1}' < +\beta_0' &\iff 1 - \beta_1' - \beta_{d-1}' && < +2\beta_0' \\
1 - \beta_0' - \beta_1'\frac{2}{d} - \beta_{d-1}' > -\beta_0' &\iff 1 - \beta_1'\frac{2}{d} - \beta_{d-1}' && > \quad 0 \\
1 - \beta_0' - \beta_1'\frac{1}{d} - \beta_{d-1}'\frac{d-1}{d} < +\beta_0' &\iff 1 - \beta_1'\frac{1}{d} - \beta_{d-1}'\frac{d-1}{d} && < +2\beta_0'
\end{aligned}
$$

combining the first and the second implies $-\frac{2d}{d-2}\beta_0' \leq \beta_1'$, while combining the second and third implies $\beta_1' \leq \frac{2d^2}{d-2}\beta_0' - \frac{d}{d-2}$. Combining these

$$-\frac{2d}{d-2}\beta_0' < \frac{2d^2}{d-2}\beta_0' - \frac{d}{d-2} \iff \beta_0' > \frac{1}{2(d+1)},$$

a contradiction to the condition at $x = 0$.

On the other hand, from Theorem 3

$$
\begin{aligned}
&x_{(1)} - \frac{1}{2(d+1)} + \frac{1}{(d+1)(d-2)}(x_{(1)} + \ldots + x_{(d)}) - \frac{(d-1)}{(d+1)(d-2)}\left((d-1)x_{(1)} + x_{(2)}\right) \\
&= \frac{1}{d+1}(x_{(1)} - x_{(2)}) + \frac{1}{(d+1)(d-2)}(x_{(3)} + \ldots + x_{(d)}) - \frac{1}{2(d+1)} \in \left[-\frac{1}{2(d+1)}, +\frac{1}{2(d+1)}\right].
\end{aligned}
$$

For Equation 18: $\left(\beta_0^\star, \beta_1^\star, \beta_{d-2}^\star, \beta_{d-1}^\star\right) = \left(\frac{1}{d^2} \quad \frac{2}{d(d-3)} \quad -1 - \frac{2}{d(d-3)} \quad 2\right)$. Suppose that there were a $\left(\beta_0' \quad \beta_1' \quad \beta_{d-2}' \quad \beta_{d-1}'\right)$ achieving a criterion less than $\beta_0^\star$. In order to scale up the proof by contradiction, we use Theorem 9.

**Theorem 9** (Carver (1922), Theorem 3). *$Ax < b$ is consistent $\iff$ $y = 0$ is the only solution for $y \geq 0$, $y^\top A = 0, y^\top b \leq 0$.*

Applying the assumed condition at $x = (0, 0, \ldots, 0)$, $(1, 1, \ldots, 1)$, $(1, 1, 1, 0, \ldots, 0)$, $(1, 1, 0, \ldots, 0)$, and $(1, 0, \ldots 0)$ imply, respectively that:[7]

$$\beta_0' < +\beta_0^\star$$
$$1 - \beta_0' - \beta_1' - \beta_{d-2}' - \beta_{d-1}' > -\beta_0^\star$$
$$1 - \beta_0' - \beta_1'\frac{3}{d} - \beta_{d-2}' - \beta_{d-1}' < +\beta_0^\star$$
$$1 - \beta_0' - \beta_1'\frac{2}{d} - \beta_{d-2}'\frac{(d+1)(d-2)}{d(d-1)} - \beta_{d-1}' > -\beta_0^\star$$
$$1 - \beta_0' - \beta_1'\frac{1}{d} - \beta_{d-2}'\frac{d-2}{d} - \beta_{d-1}'\frac{d-1}{d} < +\beta_0^\star$$

or $Ax < b$ for

$$A = \begin{pmatrix} +1 & 0 & 0 & 0 \\ +1 & +1 & +1 & +1 \\ -1 & -\frac{3}{d} & -1 & -1 \\ +1 & +\frac{2}{d} & \frac{(d+1)(d+2)}{d(d-1)} & +1 \\ -1 & -\frac{1}{d} & \frac{d-2}{d} & -\frac{d-1}{d} \end{pmatrix}, x = \begin{pmatrix} \beta_0' \\ \beta_1' \\ \beta_{d-2}' \\ \beta_{d-1}' \end{pmatrix}, \text{ and } b = \begin{pmatrix} \beta_0^\star \\ \beta_0^\star + 1 \\ \beta_0^\star - 1 \\ \beta_0^\star + 1 \\ \beta_0^\star - 1 \end{pmatrix}.$$

It is straightforward to verify that

$$y = \begin{pmatrix} 1/d \\ (d-2)/(2d) \\ (d/2 - 1) \\ (d-1)/2 \\ 1 \end{pmatrix}$$

satisfes $y^\top A = 0$ and $y^\top b = (\beta_0^\star d - 1/d) = 0$. The last equality is because $\beta_0^\star = 1/d^2$. We have demonstrated a $y \geq 0, y \neq 0$ with $y^\top b \leq 0$, thus the system is not consistent, which forms a contradiction to the supposition that there exists a $\left(\beta_0' \quad \beta_1' \quad \beta_{d-2}' \quad \beta_{d-1}'\right)$ achieving a criterion less than $\beta_0^\star$.

On the other hand, from Theorem 3

$$x_{(1)} - \frac{1}{d^2} - \frac{2}{d(d-3)}\frac{1}{d}(x_{(1)} + \ldots + x_{(d)}) + \frac{(d-2)(d-1)}{d(d-3)}2S(x; d-2, d) - \frac{2}{d}\left((d-1)x_{(1)} + x_{(2)}\right)$$
$$= \frac{2}{d^2}x_{(1)} - \frac{2}{d^2}x_{(2)} + \frac{2}{d^2}x_{(3)} - \frac{2}{d^2}\frac{1}{d-3}(x_{(4)} + \ldots + x_{(d)}) - \frac{1}{d^2} \in \left[-\frac{1}{d^2}, +\frac{1}{d^2}\right].$$

---
7

$$S(x; d-2, d) = \frac{1}{\binom{d}{d-2}}\left(\binom{d-1}{d-3}x_{(1)} + \binom{d-2}{d-3}x_{(2)} + \binom{d-3}{d-3}x_{(3)}\right)$$
$$= \frac{1}{d}\left((d-2)x_{(1)} + \frac{2(d-2)}{(d-1)}x_{(2)} + \frac{2}{(d-1)}x_{(3)}\right)$$

$\square$

For Equation 19. The idea is the same, but handling $d+1$ equations simultaneously requires more powerful notation. Let $B(d)$ be the $d-1 \times d$ matrix with $(r,c)$th element $\binom{d-c}{r-1}/\binom{d}{r}$ if $r+c \leq d+1$, and zero otherwise. We can write the condition Equation 5 simultaneously for all $r$ as

$$
S(x;d)^\top \triangleq \begin{pmatrix} S(x;0,d) \\ S(x;1,d) \\ S(x;2,d) \\ S(x;3,d) \\ \vdots \\ S(x;d-1,d) \end{pmatrix}^\top = \begin{pmatrix} 1 \\ x_{(1)} \\ x_{(2)} \\ \vdots \\ x_{(d)} \end{pmatrix}^\top \begin{pmatrix} 1 & 0_{1\times d-1} \\ 0_{d\times 1} & B(d)^\top \end{pmatrix} \in \mathbb{R}^{1\times d}. \tag{24}
$$

We have indicated the dimensionality of the zero vectors with subscripts.

Let $V(d)$ be the $d \times d+1$ matrix

$$
\begin{pmatrix}
0 & 1 & 1 & \dots & 1 & 1 \\
0 & 0 & 1 & \dots & 1 & 1 \\
\vdots & \dots & & & \vdots & \\
0 & 0 & 0 & \dots & 1 & 1 \\
0 & 0 & 0 & \dots & 0 & 1
\end{pmatrix} \tag{25}
$$

of points at which we evaluate the estimator. Let $s(d)$ be the $d+1$-dimensional vector starting and ending with $j$th element $(-1)^{j-1}$; $\mathrm{diag}(s(d))$ encodes the signs of the binding inequalities. Let

$$
A = \mathrm{diag}(s(d)) \begin{pmatrix} 1_{1\times d+1} \\ B(d)V(d) \end{pmatrix}^\top, b = 1/2^d + \mathrm{diag}(s(d)) \begin{pmatrix} 0 \\ 1 \\ 1 \\ \dots \\ 1 \end{pmatrix}.
$$

where we indicate the dimensionality of the $d+1$-dimensional vector of ones. Here

$$
y = \begin{pmatrix} \binom{d}{0} \\ \binom{d}{1} \\ \binom{d}{2} \\ \vdots \\ \binom{d}{d-1} \\ \binom{d}{d} \end{pmatrix}
$$

satisfies $y^\top A = 0$ and $y^\top b = 0$. Thus, there are no parameters achieving a criterion $< 1/2^d$, by Theorem 9.

For the other direction, let

$$\beta^\star = \begin{pmatrix} \beta_0^\star \\ \beta_1^\star \\ \vdots \\ \beta_{d-1}^\star \end{pmatrix} = -1 \times \begin{pmatrix} -1/2^d \\ (-1/2)^{d-1}\binom{d}{1} \\ (-1/2)^{d-2}\binom{d}{2} \\ \vdots \\ (-1/2)^1\binom{d}{d} \end{pmatrix}.$$

Then, by the binomial theorem[8]

$$-\begin{pmatrix} 1 & 0_{1\times d-1} \\ 0_{d\times 1} & B(d)^\top \end{pmatrix} \beta^\star = \begin{pmatrix} 1/2^d \\ 1 - 2/2^d \\ +2/2^d \\ -2/2^d \\ \vdots \end{pmatrix}.$$

Thus

$$x_{(1)} - S(x;d)^\top \beta^\star = 1/2^d + \sum_{j=1}^{d} \frac{2}{2^d}(-1)^{j+1}x_{(j)} \in \left[-\frac{1}{2^d}, +\frac{1}{2^d}\right]. \tag{26}$$

$\square$

**Theorem 10** (Fundamental perturbation analysis of $L_\infty$ optimization over $\Delta_d$). *Given a vector $\gamma \in \mathbb{R}^d$:*

$$(\gamma_{(1)} + \gamma_{(d)})/2 = \arg\min_{a\in\mathbb{R}} \max_{\lambda\in\Delta_d} |\gamma^\top\lambda - a| \tag{27}$$

$$(\gamma_{(1)} - \gamma_{(d)})/2 = \min_{a\in\mathbb{R}} \max_{\lambda\in\Delta_d} |\gamma^\top\lambda - a|. \tag{28}$$

*Proof.* The maximum absolute value element is given by $||x||_\infty = \max_{\lambda\in\Delta_d} |x^\top\lambda| = \max\{|x_{(1)}|, |x_{(d)}|\}$.

For Equation 27, for any $\lambda \in \Delta_d$, $|\gamma^\top\lambda - a| = |(\gamma - 1_d \times a)^\top\lambda|$. For any fixed $a$, the largest element (in magnitude) of $\gamma - 1_d \times a$ will be achieved at the coordinate of either the highest or lowest element of $\gamma$, and the inner optimization evaluates to

$$\max\{|\gamma_{(1)} - a|, |\gamma_{(d)} - a|\}.$$

Let $a^\star = (\gamma_{(1)} + \gamma_{(d)})/2$. Suppose that $a = a^\star + \epsilon$ for $\epsilon > 0$. Then $|\gamma_{(1)} - a| < |\gamma_{(d)} - a| = |(-\gamma_{(1)} + \gamma_{(d)})/2 - \epsilon| = (\gamma_{(1)} - \gamma_{(d)})/2 + \epsilon$, this quantity can be reduced by setting $\epsilon$ to zero. Thus the optimal $a$ is $\leq a^\star$.

Now, suppose that $a = a^\star - \epsilon$ for $\epsilon > 0$. Then $|\gamma_{(d)} - a| < |\gamma_{(1)} - a| = |(\gamma_{(1)} - \gamma_{(d)})/2 - \epsilon| = (\gamma_{(1)} - \gamma_{(d)})/2 + \epsilon$, this quantity can be reduced by setting $\epsilon$ to zero. Thus the optimal $a$ is $\geq a^\star$.

Equation 28 follows straightforwardly from plugging Equation 27 into the criterion $(\gamma_{(1)} + \gamma_{(d)})/2$ implies that the largest value of $\gamma - 1_d \times a$ is $(\gamma_{(1)} - \gamma_{(d)})/2$, and the smallest value is $(\gamma_{(d)} - \gamma_{(1)})/2$, with all other values in-between. $\square$

**Theorem 11.** *Let $L(\beta) = \begin{pmatrix} 0 & 1 & \dots 1 \end{pmatrix} - \beta^\top B(d)V(d)$, then*

$$\beta_0^\star, \beta_1^\star, \dots, \beta_{d-1}^\star \triangleq \arg\min_{\beta_0,\beta_1,\dots,\beta_{d-1}} \max_{\lambda\in\Delta_d} |L(\beta_1,\dots,\beta_{d-1})\lambda - \beta_0|$$

*satisfies $L(\beta_1^\star,\dots,\beta_{d-1}^\star) \geq 0$, $\beta^\star = \arg\min_\beta ||L(\beta)||_\infty$, and $\beta_0^\star = ||L(\beta^\star)||_\infty$.*

---

[8] $\sum_{k=0}^{n}\binom{n}{k}r^k = (1+r)^n.$

*Proof.* Note that the first column of $B(d)V(d)$ is entirely zero, so $L(\beta_1, \ldots, \beta_{d-1})_1 = 0$ for all $\beta_1, \ldots, \beta_{d-1}$. Thus, if $L(\beta_1, \ldots, \beta_{d-1}) \geq 0$, then the smallest element will be zero, and the last two assertions follow directly from Equation 27 and Equation 28.

Thus, we need only to show that at any candidate optimum $L(\beta_1, \ldots, \beta_{d-1}) \geq 0$ for all $i \iff \begin{pmatrix} \beta_1 & \beta_2 & \ldots & \beta_{d-1} \end{pmatrix} B(d)V(d) \leq 1$ for all $i$. Note that no candidate solution would have $\left( \begin{pmatrix} \beta_1 & \ldots & \beta_{d-1} \end{pmatrix} B(d)V(d) \right)_i < 0$ for any $i$ since this would result in a criterion $> 1/2$, which is trivially attainable with $\beta_1 = \ldots = \beta_{d-1} = 0$.

Suppose that at a candidate $\beta$, $m \triangleq \max_i \ (\beta^\top B(d)V(d))_i > 1$, then the minimum element of $L$ will be $1 - m < 0$, and by Equation 28, the attained criterion will then be

$$(\max_i \ L(\beta)_i - (1 - m))/2.$$

And $\max_i \ L(\beta)_i = \max(0, 1 - \min_i \ (\beta^\top B(d)V(d))_i)$. We have shown that

$$\begin{pmatrix} \beta_1 & \ldots & \beta_{d-1} \end{pmatrix} B(d)V(d) \geq 0,$$

thus $\max_i \ L(\beta/m)_i \geq \max_i \ L(\beta)_i$, and we have that

$$\max_i \ L(\beta/m)_i \leq (\max_i \ L(\beta)_i - (1 - m)) \iff (1 - m) \leq \max_i \ L(\beta)_i - \max_i \ L(\beta/m)_i.$$

We need to consider three separate cases:

1. $0 = \max_i \ L(\beta/m)_i \implies 0 = \max_i \ L(\beta)_i$ in which case the inequality holds strictly.

2. $L(\beta)_i = 0$ but $\max_i \ L(\beta/m)_i > 0$ then $\max_i \ L(\beta/m)_i = 1 - \frac{1}{m} \min_i \ (\beta^\top B(d)V(d))_i$. This holds only if $\min_i \ (\beta^\top B(d)V(d))_i > 1$, so

$$1 - m \leq -1 \times \left( 1 - \frac{1}{m} \min_i \ (\beta^\top B(d)V(d))_i \right) \iff 2 < m + \min_i \ (\beta^\top B(d)V(d))_i/m$$

which follows from the AM-GM inequality: $a > 1, b > 1 \implies (a + b/a)/2 \geq \sqrt{b} > 1$.

3. If both terms are nonzero, then

$$1 - m \leq \frac{1 - m}{m} \times \min_i \ (\beta^\top B(d)V(d))_i \iff m \geq \min_i (\beta^\top B(d)V(d))_i.$$

$\square$

**Theorem 12.** *An optimal estimator is increasing.*

*Proof.* $f$ will be increasing if and only if $\beta^\top B(d)V(d) \geq 0$.

Suppose otherwise, that for some $i \ (\beta^\top B(d)V(d))_i < 0$. Then the criterion will be $\geq (1 - \beta^\top B(d)V(d))_i)/2 > 1/2$ by Equation 27. However, by setting $\beta = 0$, a criterion of $1/2$ can always be achieved, thus $\beta$ cannot be optimal. $\square$

## B.2 Proof of Theorem 5

*Proof.* Theorem 12 shows that an optimal estimator is weakly increasing. Thus, for $\delta < \beta_0^\star$, $p \triangleq (\delta, 0, \ldots, 0) \implies f(p) \geq f(0) = \beta_0^\star > \delta = \max(p)$ and means an error of at least $\beta_0^\star - \delta$. So at $p$, the error will be at least $\epsilon$ iff $\beta_0^\star - \epsilon \geq \delta$. This is true for all $p \in [0, \delta]^d$, thus $[0, \beta_0^\star - \epsilon]^d \subseteq W(\epsilon; R) \implies \text{vol}(W(\epsilon; R)) \geq (\beta_0^\star - \epsilon)^d$.

Theorem 11 shows that $\beta_0^\star = \text{dist}(R)/2$, and the assertion is proven.

$\square$

This volume bound could be improved by including more than just the intercept in the computation of the bound. And, as one might intuit, a similar analysis holds at all vertices.

## C $L_2$ problem

For brevity in what follows, let $\kappa(d) = B(d)V(d) \in \mathbb{R}^{d-1 \times d+1}$ and let $e_{jk}$ denote the $j$-dimensional vector that is entirely zero except for the $k$th element, which is one. Then the difference between the fitted and actual values, as a function of $\lambda$ is:

$$\lambda \mapsto (1_{d+1} - e_{d+1,1})^\top \lambda - \beta_0 - \beta^\top \kappa(d)\lambda = -\beta_0 + ((1_{d+1} - e_{d+1,1}) - \kappa(d)^\top \beta)^\top \lambda. \tag{29}$$

Let $\Delta_d$ denote the $d$-dimensional unit simplex, then from Equation 29 the squared $L_2$ error is

$$\int_{\Delta_d} (\beta_0 - ((1_d - e_{d1}) - \kappa(d)^\top \beta)^\top \lambda)^2 \mathrm{d}\lambda. \tag{30}$$

To lighten the notation, we wrap this optimization problem into Theorem 13.

**Theorem 13.** *Let $\alpha_0 \in \mathbb{R}$, $\alpha \in \mathbb{R}^d$, $A \in \mathbb{R}^{d+1}$, and $\Xi \in \mathbb{R}^{d+1 \times d}$. Let $v(d) = \int_{\Delta_d} \mathrm{d}\lambda$, then*

$$\min_{\alpha_0, \alpha} \int_{\Delta_d} (\alpha_0 - (A - \Xi\alpha)^\top \lambda)^2 \mathrm{d}\lambda = v(d)A^\top \left( I - \Sigma(d)\Xi \left( \Xi^\top \Sigma(d)\Xi \right)^\dagger \Xi^\top \Sigma(d) \right) A. \tag{31}$$

*Proof.* Expanding the criterion above:

$$\alpha_0^2 v(d) - 2\alpha_0 (A - \Xi\alpha)^\top \left( \int_{\Delta_d} \lambda \mathrm{d}\lambda \right) + (A - \Xi\alpha)^\top \left( \int_{\Delta_d} \lambda\lambda^\top \mathrm{d}\lambda \right) (A - \Xi\alpha).$$

The first order criterion for optimality of $\alpha_0$ evidently requires that

$$\alpha_0 = (A - \Xi\alpha)^\top \left( \int_{\Delta_d} \lambda \mathrm{d}\lambda \right) / v(d),$$

thus, the criterion equals

$$v(d) \times (A - \Xi\alpha)^\top \left( \int_{\Delta_d} \lambda\lambda^\top \mathrm{d}\lambda / v(d) - \int_{\Delta_d} \lambda \mathrm{d}\lambda / v(d) \int_{\Delta_d} \lambda^\top \mathrm{d}\lambda / v(d) \right) (A - \Xi\alpha).$$

Write the inner term – the covariance matrix of a $\text{Dirichlet}(1, 1, \ldots, 1)$ distribution – as $\Sigma(d)$, then this weighted least squares problem is solved by

$$\alpha^\star = \left(\Xi^\top \Sigma(d)\Xi\right)^\dagger \Xi^\top \Sigma(d)A.$$

Plugging this equation into the criterion gives Equation 31. $\qquad\square$

Phrasing Equation 30 in terms of Equation 31, we have that the squared $L_2$ error of the optimal coefficients is:

$$v(d)(1_d - e_{d1})^\top \left(I - \Sigma(d)\kappa(d)^\top \left(\kappa(d)\Sigma(d)\kappa(d)^\top\right)^\dagger \kappa(d)\Sigma(d)\right)(1_d - e_{d1}). \tag{32}$$

The hat matrix $\Sigma(d)\kappa(d)^\top \left(\kappa(d)\Sigma(d)\kappa(d)^\top\right)^\dagger \kappa(d)\Sigma(d)$ has rank $d - 1$, thus $(1_d - e_{d1})$ cannot possibly lie in the nullspace of the projection operator, and we have a strictly positive error. We skip deriving the exact expressions as a function of $d$ and note that the same analysis could be straightforwardly conducted constraining different coefficients to equal zero.

# D Distribution of optimal error

In this section, we derive the distribution of the optimal residual.

## D.1 A foundational result

Weisberg (1971) gives this result:

> Let $x_{(0)}, x_{(1)}, x_{(2)}, \ldots, x_{(d)}, x_{(d+1)}$ be the order statistics from $x$ sampled uniformly on the $d$-dimensional unit cube with the convention that $x_{(0)} = 0, x_{(d+1)} = 1$.
>
> Let $\alpha_j \geq 0, j = 1, \ldots, d$ be positive weights, and let $a_j = \sum_{k=j}^{d} \alpha_k$, so $a_1 \geq a_2 \geq \ldots \geq a_d$. Let $c_1 > c_2 > \ldots > c_s$ be the unique (strictly ordered) values of $a_k$, and $k_\ell$ denotes the number of $a_j$ that $c_\ell$ covers.
>
> Then
>
> $$\Pr\left[\sum_{j=1}^{d} x_{(j)}\alpha_j \leq z\right] = 1 - \sum_{i=1}^{r} \frac{\mathrm{d}^{k_i-1}}{\mathrm{d}x^{k_i-1}} \left.\frac{(x-z)^d}{\prod_{\ell \neq i}(x - c_\ell)}\right|_{x=c_i}, \tag{33}$$
>
> where $r$ is the largest $i$ where $z \leq c_i$.

Matsunawa (1985) also gives a characteristic function-based analysis.

In our application, the weights in the linear combination are not nonnegative. Happily, there is a reduction to this case presented by Diniz et al. (2002) (modernized from Dempster & Kleyle (1968)):

> Let $\alpha_j$ be as before, though not necessarily nonnegative this time, and as before let $a_j = \sum_{k=j}^{d} \alpha_k$. Let $a^0 = (0, a_1, \ldots, a_d)$ be $a$ prepended with a zero and let $\sigma$ be the permutation of the $\{1, \ldots, d+1\}$ so that $a^0_{\sigma(1)} \geq \ldots \geq a^0_{\sigma(d+1)}$. Then
>
> $$\Pr\left[\sum_{j=1}^{d} x_{(j)}\alpha_j \leq z\right] = \Pr\left[\sum_{j=1}^{d} x_{(j)}(a^0_{\sigma(j)} - a^0_{\sigma(j+1)}) \leq z - a^0_{\sigma(d+1)}\right]. \tag{34}$$

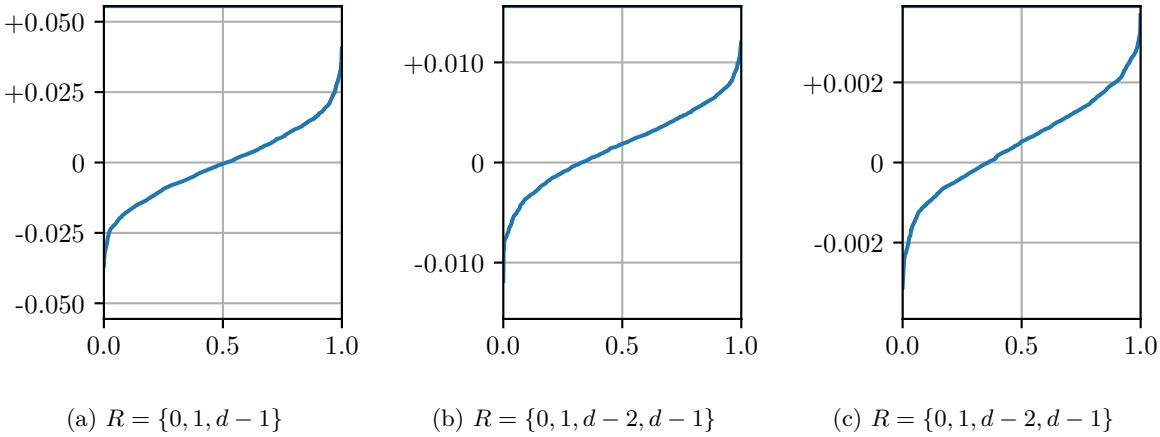

(a) $R = \{0, 1, d-1\}$      (b) $R = \{0, 1, d-2, d-1\}$      (c) $R = \{0, 1, d-2, d-1\}$

Figure 7: Simulated empirical cumulative distribution functions for $x_{(1)} - f_R^\star(x)$ over 1500 uniform random points in the unit cube, for three $R$. The $y$ axis is scaled between $\pm$ the maximum $L_\infty$ error.

Basically: you include zero with the $a$ as calculated previously, sort them, then use the previous logic with the differences (guaranteed to be nonnegative), and a modified argument, you get the same probability. In this equation, some differences may be zero, and that is covered in Equation 33 with zero $\alpha_j$s.

Actually evaluating the derivatives in Equation 33 is not too difficult – there is an efficent recursive formulation – but is also not totally straightforward. Thus, for brevity we do not go into too much more detail and instead simply observe that the cumulative distribution function is a continuously differentiable function of its input. To give an idea of the distribution, Figure 7 plots the empirical distribution for 1500 points.

### D.2 The residual as a linear combination of order statistics

The optimal residual is a linear combination of order statistics, from Equation 13 and Equation 5:

$$
\begin{aligned}
f_R^\star(x) &= \sum_{r \in R} \beta_r^\star S(x; r, d) \\
&= \sum_{r \in R} \beta_r^\star \frac{1}{\binom{d}{r}} \sum_{j=1}^{d-r+1} \binom{d-j}{r-1} x_{(j)} \\
&= \sum_{k=1}^{d} a_k^\star x_{(k)} \text{ where } a_k^\star = \sum_{r \in R \cap \{0, \ldots, d-k+1\}} \frac{\beta_r^\star}{\binom{d}{r}} \binom{d-k}{r-1}.
\end{aligned}
$$

Thus, the optimal residual is a linear function of the order statistics:

$$
x_{(1)} - f_R^\star(x) = (1 - a_1^\star) x_{(1)} - \beta_0^\star - \sum_{r \in R \backslash \{0,1\}} a_r^\star x_{(r)}.
$$

The cumulative distribution of this quantity then follows from plugging these coefficients into Equation 34. This distribution is a a polynomial, so continuous, thus the probability that it takes any discrete value is zero.

## E  Bounding the error and complexity of a general $R$-estimator

In this section, we present a method for computing nontrivially tight lower bounds on the error from a general $R \subseteq \{0, 1, 2, \ldots, d-1, d\}$ estimator. For any set of points $P \subseteq [0,1]^d$, we have that

$$||m - \max||_\infty \geq \max_{x \in P} |m(x) - \max(x)| \implies$$
$$\min_{m \in \mathcal{M}_d^s(R)} ||m - \max||_\infty \geq \min_{m \in \mathcal{M}_d^s(R)} \max_{x \in P} |m(x) - \max(x)|.$$

Apply this with $P$ equal to $d + 1$ corners of the unit cube containing zero, one, two, etc. ones:

$$||m - \max||_\infty \geq \min_{m \in \mathcal{M}_d^s(R)} \max \{|m((0 \quad 0 \quad \ldots \quad 0 \quad 0))|,$$
$$|m((1 \quad 0 \quad \ldots \quad 0 \quad 0)) - 1|,$$
$$|m((1 \quad 1 \quad \ldots \quad 0 \quad 0)) - 1|,$$
$$\ldots$$
$$|m((1 \quad 1 \quad \ldots \quad 1 \quad 0)) - 1|,$$
$$|m((1 \quad 1 \quad \ldots \quad 1 \quad 1)) - 1|\}.$$

Finally, we write this as the convex optimization problem, in $2(d+1)$ constraints, and $1 + |R|$ variables, using a a standard trick for rewriting $L_\infty$ optimization (see Boyd & Vandenberghe (2004)).

$$\min_{g, \beta_0, (\beta_r, r \in R)} g \text{ subject to } \left|\beta_0 + \sum_{r \in R \setminus \{0\}} \beta_r S((0 \quad 0 \quad \ldots \quad 0 \quad 0); r, d)\right| \leq g,$$

$$\left|\beta_0 - \sum_{r \in R \setminus \{0\}} \beta_r S((1 \quad 0 \quad \ldots \quad 0 \quad 0); r, d) - 1\right| \leq g,$$

$$\left|\beta_0 - \sum_{r \in R \setminus \{0\}} \beta_r S((1 \quad 1 \quad \ldots \quad 0 \quad 0); r, d) - 1\right| \leq g,$$

$$\ldots$$

$$\left|\beta_0 - \sum_{r \in R \setminus \{0\}} \beta_r S((1 \quad 1 \quad \ldots \quad 1 \quad 0); r, d) - 1\right| \leq g,$$

$$\left|\beta_0 - \sum_{r \in R \setminus \{0\}} \beta_r S((1 \quad 1 \quad \ldots \quad 1 \quad 1); r, d) - 1\right| \leq g.$$

This computation scales well, essentially linear programs such as this can be simply solved on a desktop computer using standard software for thousands of variables and constraints.

## F    Implementing an $R$ estimator as a feedforward network

In this section, we show how to cast a general $R$ estimator as the forward pass of a feedfoward network. This analysis is necessary to give a benchmark against which to compare stochastic gradient descent fitting. The code, in idiomatic PyTorch and accompanied by extensive test cases, is available at `https://github.com/idiap/benefits-of-max-pooling`.

First, we describe a concept called the $R$-mapping, then we describe an algorithm for computing $R$-mappings, and then we show how to use an $R$-mapping to construct a feedforward network that is an $R$-estimator.

### F.1 $R$-mapping definition and motivation

An *R-mapping* describes an $R$-estimator as a sequence of pairwise maxes. For $d \in \mathbb{N}$ and $r \in \{1, 2, \ldots, d\}$ let $C(r, d) = \{\{1, 2, \ldots, r\}, \{1, 3, \ldots, r + 1\}, \ldots, \{d - r, \ldots, d\}\}$ denote the set of size $\binom{d}{r}$ of all subsets of $(1, 2, \ldots, d)$ of size $r$.

**Definition 1.** *An R-mapping for $R \subseteq \{0, 1, 2, \ldots, d\}$ is a sequence of sets $t_1, t_2, \ldots, t_s$ with $s \leq \lceil \log_2 d \rceil$ satisfying:*

- *for all $r \in R, r > 0$ there is some $j$ such that $C(r, d) \subseteq t_j$, and*

- $t_{j+1} \subseteq \{\tau_1 \cup \tau_2 : \tau_1, \tau_2 \in t_j\}.$

At a high level: Each element of an $R$-mapping corresponds to a set of indices into the input, and at the $j$th layer computes the max of the input over all indices in $t_j$. The first defining characteristic of an $R$-mapping ensures that the indices permit the evaluation of all required subpool max averages. And the second condition insures that a feedforward network can compute the maxes over the implied indices.

To simplify the subsequent discussion, we hereafter assume that $\{0, 1\} \subseteq R$ for all $R$. Since the first two terms are trivially uncomplicated – a constant bias, and the grand mean of the inputs are linear features – this assumption is without any loss of generality and could be easily relaxed.

### F.2 Computing $R$-mappings

In this section, we show how to compute an $R$-estimator. We say that $R$ is *adequate* if $\max(R) \leq 2 \times \max\left(R \cap [0, 2^{\lfloor \log_2(\max(R)-1) \rfloor}]\right)$. If $R$ is adequate then it possible to form the greatest remaining term from pairwise maxes of terms that are a lower power of two – a condition necessary to enforce the second condition in Definition 1. If $R$ is not adequate, then it can be made adequate by appending an additional term.

Let $\widetilde{R} = a(R)$, where $a : \{0, \ldots, d\} \mapsto \{0, \ldots, d\}$ is defined recursively as:

$$
a(R) = \begin{cases} R & \text{if } R = \{0, 1\} \\ \{\max(R)\} \cup a(R \backslash \{\max(R)\}) & \text{if } R \text{ is adequate} \\ \{\max(R)\} \cup a(R \backslash \{\max(R)\}) \cup \{\lceil \max(R)/2 \rceil\} & \text{otherwise.} \end{cases} \tag{35}
$$

The third case covers the situation where it would not be possible to compute an $R$-mapping out of terms in $R$, and so an additional term is appended. For example, $a(\{0, 1, 2, 5, 6\}) = \{0, 1, 2, 3, 5, 6\}$: 3 has been appended since it is impossible to compute the maxes of five and six terms using only pairwise maxes ($r = 2$) of pairs of variables. $a(R)$ essentially reduces its argument by one term with each recursive call, and thus it is fast and straightforward to evaluate.

By construction, every truncation of $\tilde{R}$ is adequate, thus for every $\tilde{r} \in \widetilde{R}$ with $\tilde{r} > 1$ there exists an $\tilde{r}' \in \widetilde{R}$ with $\tilde{r}' \geq \tilde{r}/2$. This means that $C(1, d), C(2, d), \cup_{r \in R \cap [3,4]} C(r, d), \ldots, \cup_{r \in R \cap [d/2, d]} C(r, d)$, is a $R$-mapping, however if $R \subsetneq \tilde{R}$, then there will be smaller $R$-mappings since it is possible to skip the computation of some terms in $C(r, d)$. For example, continuing our example above, there are 56 subsets of size 3 of $d = 8$ values, but 36 terms of length 3 can be combined to form all subsets of size 5 and 6.

For a vector $A \in \mathbb{R}^\ell$ with $\ell > k$, let $\text{SPLIT}_k : \mathbb{R}^\ell \to \mathbb{R}^k \times \mathbb{R}^k$ be the function that splits its argument into the first and last $k$ elements: $\text{SPLIT}_k(A) = (A_{(1)}, \ldots, A_{(k)}), (A_{(\ell-k)}, \ldots, A_{(\ell)})$. And let $\text{FLATSPLIT}_k$ be the set-valued function that applies $\text{SPLIT}$ to each of its inputs and collects the outputs into a single set

$$
\begin{aligned}
&\text{FLATSPLIT}_k(\{A^1, A^2, \ldots, A^n\}) \\
&= \{(A^1_{(1)}, \ldots, A^1_{(k)}), (A^2_{(\ell-k)}, \ldots, A^2_{(\ell)}), \ldots, (A^n_{(1)}, \ldots, A^n_{(k)}), (A^n_{(\ell-k)}, \ldots, A^n_{(\ell)})\}.
\end{aligned}
$$

---

**Algorithm 1** Computation of an $R$-mapping

---

    **Input:** $R \subseteq \{0, 1, \ldots, d\}$
    **Output:** $R$-mapping suitable for evaluating $f_R^\star$
    $\tilde{R} \leftarrow a(R)$ // Augment $R$ with needed terms
    $s = \lceil \log_2(\max R) \rceil$
    $R_j \leftarrow R \cap \{i : i \in \mathbb{N}, \lceil \log_2 i \rceil = j\}$ for $j = 1, 2, \ldots, s$ // group $R$ by power of 2
    $\tilde{R}_j \leftarrow \tilde{R} \cap \{i : i \in \mathbb{N}, \lceil \log_2 i \rceil = j\}$ for $j = 1, 2, \ldots, s$ // group $\tilde{R}$ by power of 2
    **for** $i = 0, 1, \ldots, s - 1$ **do**
      $j \leftarrow s - i - 1$ // backward index
      **if** $i = 0$ **then**
        $N_j \leftarrow \emptyset$
      **else**
        $N_j \leftarrow t_{j+1}$ // terms needed for subsequent layer
      **end if**
      $X_j \leftarrow \{C(k, r, d) : k = 1, \ldots, \binom{d}{r}, r \in R_j\}$ terms needed for this layer
      $Y_j \leftarrow X_j \cup N_j$
      $k_j \leftarrow \max(\tilde{R}_j)$ // tuple width in this mapping term
      $t_j \leftarrow \text{FLATSPLIT}_{k_j}(Y_j)$
    **end for**
    **Return** $t_1, t_2, \ldots, t_s$

---

Algorithm 1 gives one approach to compute an $R$-mapping that improves upon naïvely computing all subpool maxes of order $\tilde{r} \in \tilde{R}$: The idea is to include only those terms in $\tilde{R}$ only minimally in order to support the computation of subsequent terms.

### F.3    $R$-mappings to neural network $R$-estimators

An $R$-mapping, as introduced in subsection F.2, is a sequence of sets that is defined so that each element of a constituent set is associated with a pair of elements in the previous set. Thus, it is well-suited to compute pairwise maxes via a simple linear-ReLU-linear block as shown in Equation 3. Computing the average of all subpooled values is of course a linear operation.

An important book-keeping challenge with this approach is to enable the network to convey the average of low-order subpool maxes through the network. To do this, we append to the network a "memory" – additional neurons which carry forward values computed earlier in the network via identity mappings (propagated through ReLUs via $x = \text{ReLU}(+x) - \text{ReLU}(-x)$) through until the end – a layer of width $|R|$, containing all needed subpool max averages. Finally, then, these values are aggregated according to $\beta$.

Our code is written in three stages: (1) compute a base network consisting of the linear layers implied by Equation 3, then (2) append the subpool averages and their attendant memory neurons, and finally (3) aggregate the penultimate layer value with the coefficients. One helpful trick to developing this logic is to leave each step above as consecutive linear layers, then once everything is complete, to fuse them all together.

This approach perhaps does not result in the smallest possible feedforward network that could implement an $R$-estimator but it is relatively simple to code, fast to run, and the architecture is very descriptive of the logic the network implements. It seems unlikely that a large improvement on this general scheme is possible, though we do not attempt to prove this speculation.

## G    Additional experiments

In this section, we repeat Figure 2 for configurations different than the one described in subsection 5.1 in exactly one way, described in the caption. Across Figure 8, Figure 9, Figure 10, Figure 11, Figure 12, and Figure 13 the basic pattern of quickly falling error that levels out around $\mu = 1$ for all three models is repeated.

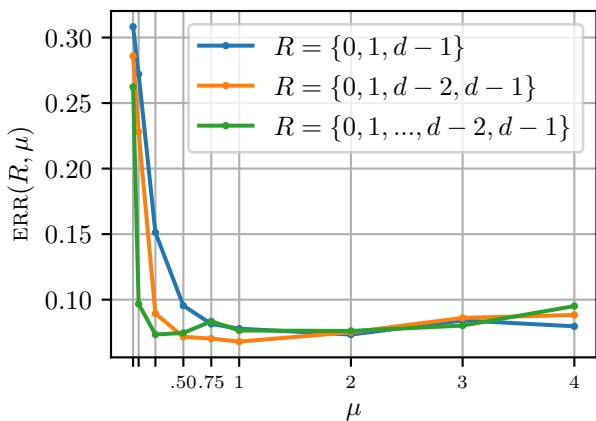

Figure 8: Figure 2 with Kaiming initialization for weights and small positive constant for bias.

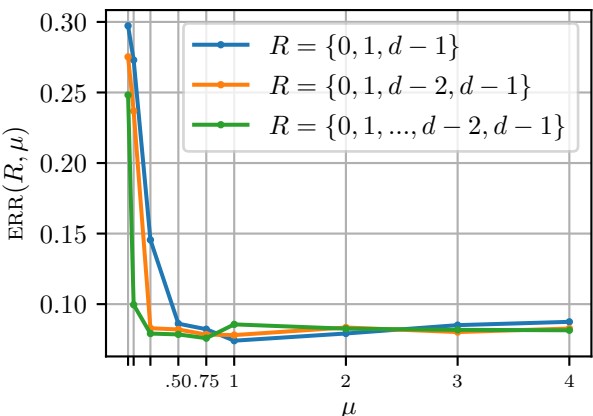

Figure 9: Figure 2 with Xavier initialization for weights and small positive constant for bias.

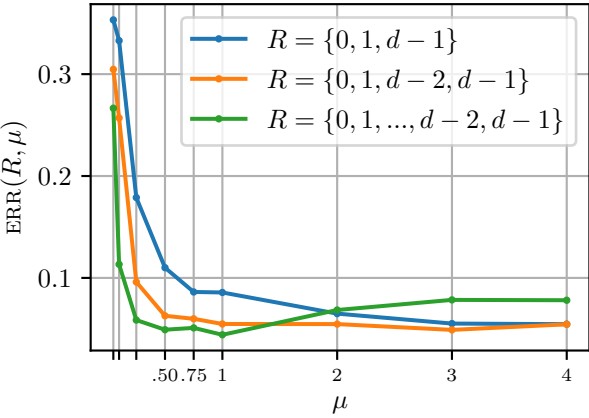

Figure 10: Figure 2 with $L_\infty$ criterion replaced with $L_2$ criterion.

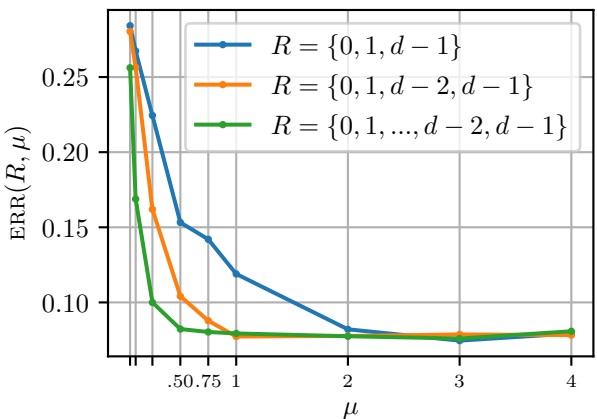

Figure 11: Figure 2 with Adam optimizer replaced with AdamW optimizer (PyTorch default parameters).

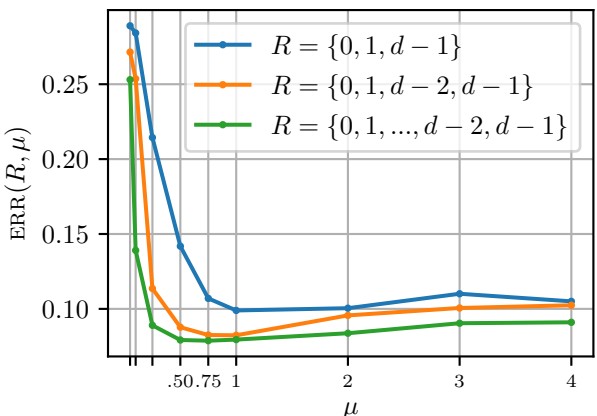

Figure 12: Figure 2 with pseudorandom uniform data replaced with Sobol sequence.

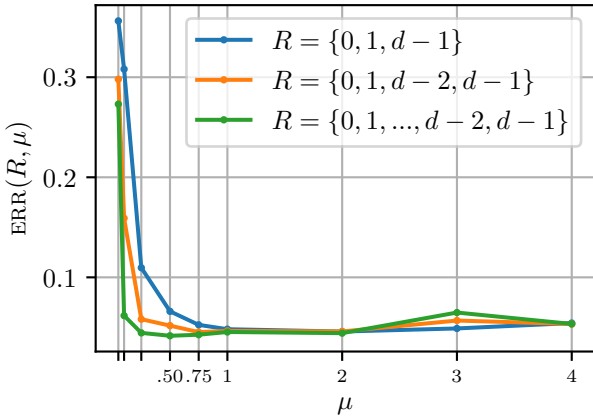

Figure 13: Figure 2 with pseudorandom uniform data replaced with Dirichlet$(1, \ldots, 1)$ data.

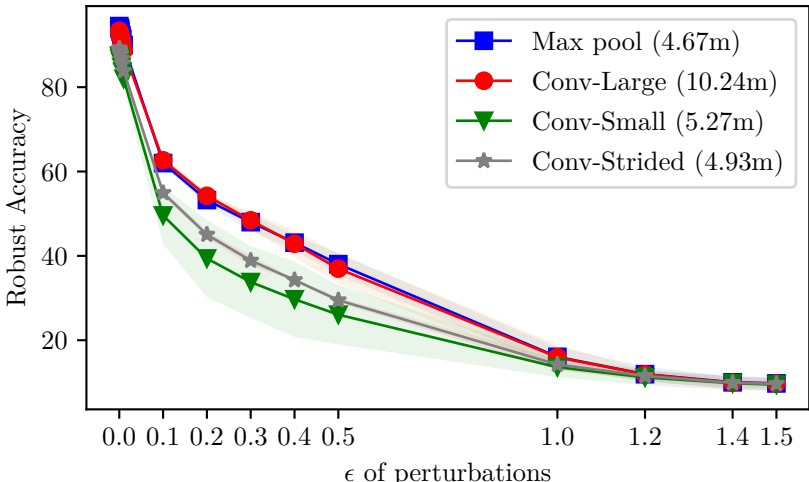

Figure 14: Effect of perturbations with $L_\infty \leq \epsilon$ on the accuracy of the ResNet model variants on the CIFAR10 dataset. Mean accuracy and $\pm 1.96$ standard deviation over the course of three runs are depicted. The legend indicates the number of parameters of each model in parentheses. The models omitting max pool pay a price in either (robust) accuracy or model complexity (in terms of parameters).

## H  Analysis

We have strictly looked at the accuracy implications of approximating the max function in isolation. In this short and speculative section, we examine briefly the practical takeaways for adversarial robustness. The code to implement this analysis is at `https://github.com/nik-dim/maxpooling`.

### H.1  Adversarial robustness

Our hypothesis is that max pooling can be more robust than strided convolution and ReLU nonlinearity, since genuine max pooling admits only a single direction along which features can change – the max. ReLU, by contrast, can be moved with only low correlation changes, and a random perturbation will in general change the output.

Our experiments corroborate this intuition; omitting max pooling results in lower robust accuracy in several different model classes, ranging from simple Convolutional Neural Networks to ResNets. Our experimental approach is to adversarially attack models with and without max pooling. We use the Fast Gradient Sign Method by Goodfellow et al. (2015). Specifically, starting from a model incorporating max pool layers, we replace them with strided convolution + ReLU. We examine four different models and report the robust accuracy in Figure 14 on the CIFAR10 dataset (Krizhevsky (2009)). The Max pool model is from Page (2018) and includes four max pool layers. The baselines make the following modifications: Conv-Small and Conv-Large replace the max pool layers with a convolutional one, with kernel size one and three, respectively. The Conv-Strided model uses a strided convolution in lieu of the max pool layer and the convolution that precedes it.

We performed the computation on an internal computation cluster of Tesla V100-SXM2-32GB GPUs. All experiments presented here can be done in less than seven Tesla V100 days. All software and data are standard academic tools and present no licensing issues.

The experiments are developed in PyTorch Paszke et al. (2017) and PyTorch Lightning Falcon (2019) with the help of the foolbox Rauber et al. (2017) library for the adversarial attack. We use a ResNet variant in our experiments Page (2018). We use the publicly available datasets MNIST (LeCun et al., 2010) and CIFAR10 Krizhevsky (2009).

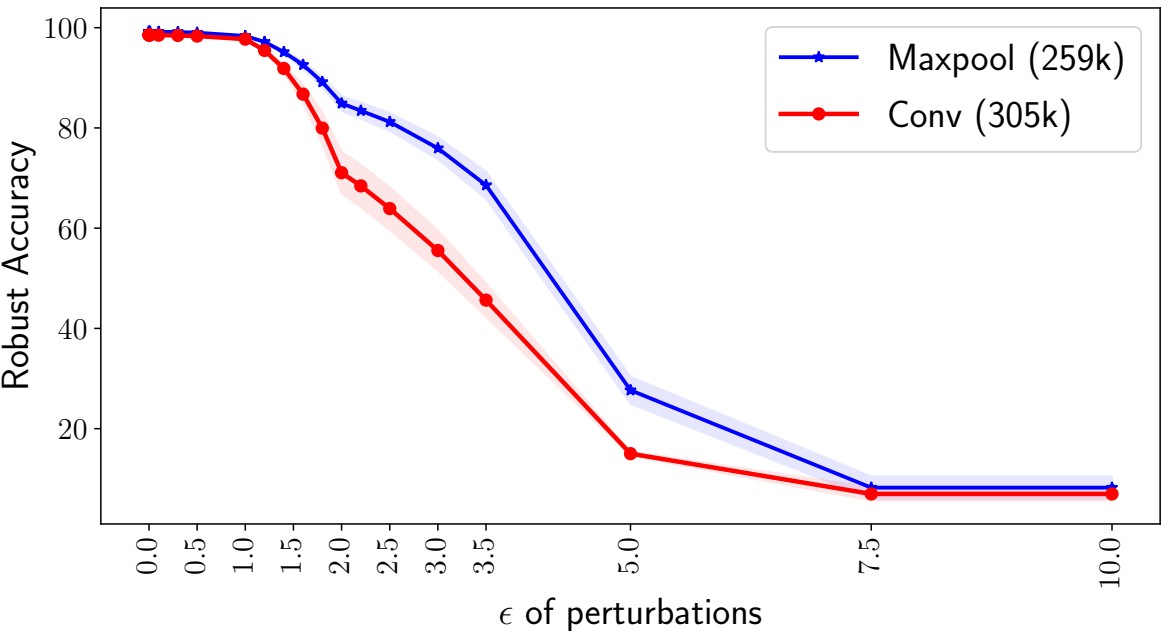

Figure 15: Effect of perturbations with $\ell_\infty \le \epsilon$ on the accuracy of the LeNet model variants on the MNIST dataset.

The experimental results showcase a tradeoff between model complexity (in terms of number of parameters) and robust accuracy. The Max pool model is more adversarially robust than the baselines Conv-Small and Conv-Strided. This effect is further highlighted for larger perturbations $\epsilon$. An exception to this trend lies in the Conv-Large model which is able to match the robust accuracy of the Max pool model, but requires more than twice as many parameters.

# I  Experimental Details

In each experiment, we create a model incorporating max pool layer(s). Then, the network is modified by replacing each max pool layer with a trainable variant, ensuring that the output of the original layer and the modified one have the same shape.

The legend of each figure presents the name of the model variant as well as the number of trainable parameters in parentheses. The width of the lines is proportional to the number of parameters in the model. The max pool model variant is always depicted in blue.

## I.1  Experimental configurations

**LeNet experiment on MNIST**   We train two convolutional neural networks (CNNs) on the digit classification dataset MNIST. The results are shown in Figure 15.

The first model, in blue, has 259 106 trainable parameters and consists of two convolutional layers, with 32 and 64 channels. Their kernel size is equal to five. Both layers are succeeded by a two-dimensional max pool with kernel size, stride and padding equal to three, two and one, respectively. The network is completed with two fully-connected layers of 1024 and 200 neurons, leading to the output of ten logits. The second model, in red, has 305 282 trainable parameters and has the same structure as the previous model. However, the max pool layers are replaced by a convolutional layer with the same number of input channels as the output of the preceding convolutional layer, hence the increase in parameters. Both models are trained for ten epochs of Stochastic Gradient Descent with learning rate and momentum equal to 0.01 and 0.9, respectively.

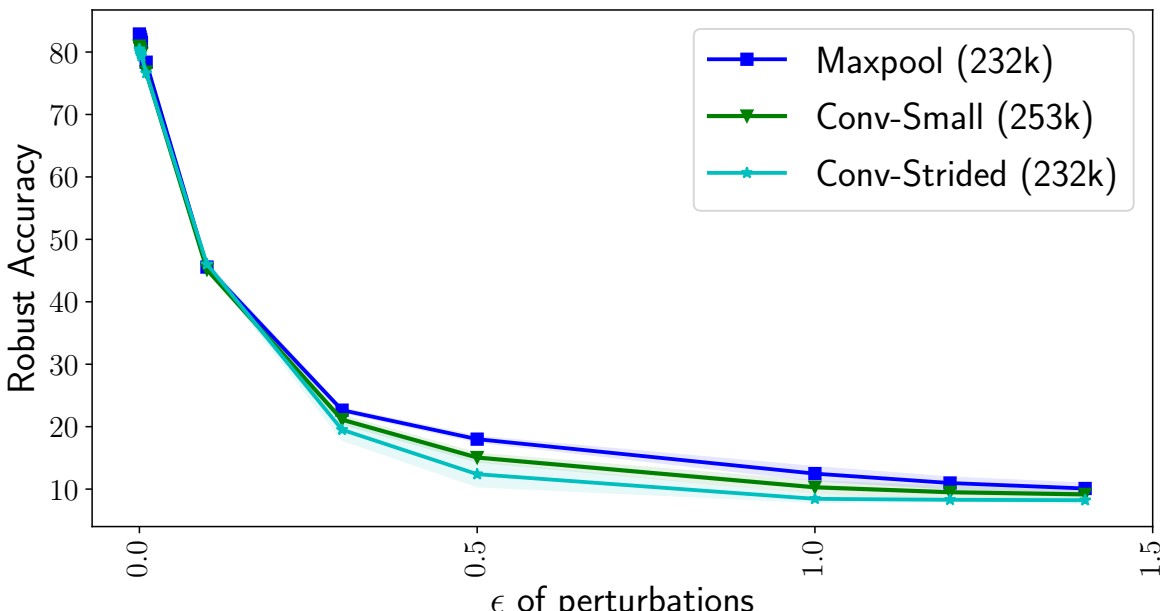

Figure 16: Effect of perturbations with $\ell_\infty \leq \epsilon$ on the accuracy of the LeNet model variants on the CIFAR10 dataset.

**LeNet experiment on CIFAR10** A similar experiment is performed on the more challenging CIFAR10 dataset. The results are shown in Figure 16.

The max pool model, in blue, has 232 162 parameters and consists of three convolutional layers of 32, 64, 128 channels, respectively. Again, each of these layers is succeeded by max pool module identical to the MNIST experiment. The convolutional variant, depicted in green, has 253 890 parameters has a similar modification as before, i.e. the max pool layer is replaced by a convolutional one with kernel size, stride and padding identical to the corresponding max pool layer. Finally, the strided variant, in gray, has the same number of parameters as the original max pool model. In this case, we replace the block of convolution and max pool with a strided convolution of stride equal to two. Hence, this modification does not incur an increase in number of parameters, while maintaining the same output shape at all intermediate steps. The models are trained for 100 epochs of SGD with learning rate and momentum equal to 0.01 and 0.9 respectively. The learning rate is decayed by a parameter $\gamma = 10$ in epochs 50, 70 and 90. The batch size is 128.

**ResNet experiment on CIFAR10** We use the ResNet variant proposed in Page (2018). This model consists of a preparatory whitening layer, three residual blocks and a classifier layer. First, the preparatory layer has two convolutions and Ghost Batch Normalization Hoffer et al. (2017). The three residual blocks have identical structure, with the exception of the number of channels in the convolutional layer; the channels are doubled with each layer, from 64 to 128 to 256. Each of these layers consists of two blocks: the first one has a convolutional layer, a max pool layer with kernel size and stride equal to two and Ghost Batch Norm, while the second block employs a residual connection with similar structure as the previous block modulo the max pool. Finally, the classifier layer is comprised of a max pooling layer of kernel size and stride equal to four and a fully connected layer resulting in ten outputs.

Overall, the Maxpool model, in blue, has four maxpooling layers and 4 666 265 parameters. The large convolutional model, depicted in red, has 10 238 233 parameters and replaces the Maxpool layers by convolutional ones with the same kernel size, stride and padding. The small convolution variant, depicted in green, has 5 273 881 parameters and the kernel size is set to one for all maxpooling replacements. Finally, the strided model, depicted in gray, has 4 928 921 parameters and. for each residual layer, the pair of convolution and maxpooling is replaced by a strided convolution, while the maxpooling layer preceding the fully connected

one is replaced with a convolutional layer of kernel size equal to one. The stride remains equal to four, as in the corresponding maxpooling layer.

The models are trained for 50 epochs using float 16 precision. The learning rate follows a piecewise linear schedule; starting at zero the learning rate linearly increases to 0.4 until the fifth epoch and then linearly decays to zero until the final epoch.

### I.2   Analysis

Our objective lies in showing the superior adversarial robustness of models incorporating max pooling. In each experiment, we use a max pooling model, drawn from widely used neural networks such as LeNet and ResNet. We modify the original model to produce comparisons. Specifically, the modifications simply replace the maxpooling layer with a convolutional layer, or the pair of convolutional layer and maxpooling (which traditionally come succession) with a strided convolution. In both cases, the new layer produces outputs of the same shape as the original layer, lending itself to an one-to-one comparison in terms of performance on (robust) accuracy. It is important to note that the first modification results in an increase in the number of trainable parameters. Subsequently, we perform an adversarial attack, FGSM in our case, to illuminate the adversarial robustness properties of each model. A common theme of all experiments is that the exclusion of the maxpooling layer results in a tradeoff between (robust) accuracy and model complexity.

First, in the MNIST experiment both variants reach similar levels of performance on the clean accuracy ($\epsilon = 0$); the max pool variant achieves $99.19 \pm 0.06$ while the convolutional model $98.54 \pm 0.14$. This is not surprising given the low difficulty of the dataset. Nevertheless, the model *with* maxpooling is characterized by strictly higher adversarial robustness, since the difference in performance heightens for larger $\epsilon$. In the CIFAR10 experiment, the observations are similar in nature; replacing maxpooling with a trainable layer renders the model more susceptible to adversarial perturbations. However, the modified models do not exhibit the same level of clean accuracy, despite the increase in model complexity. Specifically, the mean clean accuracies (over 3 runs with different random seeds) of the max pool, small convolutional and strided convolution models are $82.89 \pm 0.08\%$, $80.94 \pm 0.26\%$ and $80.45 \pm 0.66\%$, respectively. It is important to note that the LeNet architecture does not achieve state-of-the-art results on any of the model variants presented. However, it serves as a direct comparison with the previous experiment. Finally, the ResNet experiment perhaps illuminates the tradeoff more clearly. Figure 14 (see main text) presents a dichotomy due to the exclusion of the max pool layer; the practitioner should choose between model complexity (measured in number of trainable parameters and, by extension, training and inference times) and (robust) accuracy. The large convolution model achieves a clean accuracy of $93.37 \pm 0.14\%$ compared to $94.49 \pm 0.20\%$ of the original model and is able to match its robust accuracy for different $\epsilon$, while using more than double the parameters. The other two variants, however, have lowest clean accuracies ($89.22 \pm 0.09\%$ for the strided model and $87.46 \pm 0.99\%$ for the small convolutional) and present a faster deterioration in adversarial robustness.

### I.3   Detailed Results

For completeness, we present the experimental results in tabular form. The experiments were repeated three times (with different random seeds) and the mean $\pm$ standard deviation is reported.

Table 2: Detailed results on MNIST.

| $\epsilon$ | Maxpool | Conv |
|---|---|---|
| 0.000 | $99.19 \pm 0.06$ | $98.54 \pm 0.14$ |
| 0.001 | $99.19 \pm 0.06$ | $98.54 \pm 0.14$ |
| 0.002 | $99.19 \pm 0.06$ | $98.54 \pm 0.14$ |
| 0.003 | $99.19 \pm 0.06$ | $98.54 \pm 0.14$ |
| 0.010 | $99.19 \pm 0.06$ | $98.54 \pm 0.14$ |
| 0.100 | $99.18 \pm 0.05$ | $98.50 \pm 0.16$ |
| 0.300 | $99.14 \pm 0.04$ | $98.45 \pm 0.16$ |
| 0.500 | $99.00 \pm 0.07$ | $98.33 \pm 0.11$ |
| 1.000 | $98.34 \pm 0.09$ | $97.72 \pm 0.07$ |
| 1.200 | $97.16 \pm 0.08$ | $95.47 \pm 0.15$ |
| 1.400 | $95.14 \pm 0.38$ | $91.86 \pm 0.97$ |
| 1.600 | $92.59 \pm 0.72$ | $86.74 \pm 1.81$ |
| 1.800 | $89.16 \pm 1.05$ | $79.97 \pm 3.15$ |
| 2.000 | $84.90 \pm 1.46$ | $71.07 \pm 4.14$ |
| 2.200 | $83.44 \pm 1.62$ | $68.43 \pm 4.22$ |
| 2.500 | $81.20 \pm 1.77$ | $63.91 \pm 4.27$ |
| 3.000 | $75.95 \pm 2.20$ | $55.55 \pm 3.99$ |
| 3.500 | $68.58 \pm 2.69$ | $45.65 \pm 3.32$ |
| 5.000 | $27.67 \pm 2.71$ | $15.02 \pm 0.46$ |
| 7.500 | $8.24 \pm 2.22$ | $6.98 \pm 1.31$ |
| 10.000 | $8.24 \pm 2.22$ | $6.98 \pm 1.31$ |

Table 3: Detailed results on CIFAR10 with LeNet.

| $\epsilon$ | Maxpool | Conv-small | Strided |
|---|---|---|---|
| 0.000 | $82.89 \pm 0.09$ | $80.94 \pm 0.27$ | $80.45 \pm 0.67$ |
| 0.001 | $82.42 \pm 0.11$ | $80.51 \pm 0.19$ | $80.08 \pm 0.60$ |
| 0.002 | $82.00 \pm 0.08$ | $80.09 \pm 0.20$ | $79.70 \pm 0.58$ |
| 0.003 | $81.56 \pm 0.13$ | $79.68 \pm 0.21$ | $79.31 \pm 0.66$ |
| 0.010 | $78.37 \pm 0.03$ | $76.80 \pm 0.17$ | $76.64 \pm 0.30$ |
| 0.100 | $45.56 \pm 0.54$ | $45.16 \pm 0.41$ | $46.02 \pm 0.30$ |
| 0.300 | $22.64 \pm 0.32$ | $21.12 \pm 0.68$ | $19.51 \pm 1.61$ |
| 0.500 | $18.00 \pm 0.42$ | $15.05 \pm 0.65$ | $12.39 \pm 1.96$ |
| 1.000 | $12.48 \pm 1.08$ | $10.29 \pm 0.96$ | $8.44 \pm 0.52$ |
| 1.200 | $10.97 \pm 0.90$ | $9.50 \pm 0.90$ | $8.28 \pm 0.21$ |
| 1.400 | $10.10 \pm 0.81$ | $9.15 \pm 0.75$ | $8.22 \pm 0.46$ |

Table 4: Detailed results on CIFAR10 with ResNet.

| $\epsilon$ | Maxpool | Conv-Large | Conv-Small | Strided |
|---|---|---|---|---|
| 0.000 | $94.49 \pm 0.20$ | $93.37 \pm 0.14$ | $87.46 \pm 0.99$ | $89.22 \pm 0.09$ |
| 0.001 | $94.03 \pm 0.26$ | $92.95 \pm 0.16$ | $87.01 \pm 1.05$ | $88.79 \pm 0.14$ |
| 0.002 | $93.67 \pm 0.15$ | $92.58 \pm 0.08$ | $86.51 \pm 1.10$ | $88.27 \pm 0.15$ |
| 0.003 | $93.18 \pm 0.11$ | $92.14 \pm 0.04$ | $85.95 \pm 1.07$ | $87.83 \pm 0.18$ |
| 0.005 | $92.32 \pm 0.25$ | $91.27 \pm 0.08$ | $84.81 \pm 1.15$ | $86.72 \pm 0.26$ |
| 0.007 | $91.46 \pm 0.24$ | $90.22 \pm 0.12$ | $83.68 \pm 1.18$ | $85.62 \pm 0.25$ |
| 0.010 | $89.99 \pm 0.28$ | $88.72 \pm 0.09$ | $82.00 \pm 1.13$ | $83.89 \pm 0.29$ |
| 0.100 | $62.00 \pm 0.27$ | $62.66 \pm 0.40$ | $49.58 \pm 3.48$ | $54.93 \pm 0.38$ |
| 0.200 | $53.24 \pm 0.64$ | $54.24 \pm 0.38$ | $39.39 \pm 4.68$ | $45.05 \pm 0.42$ |
| 0.300 | $48.00 \pm 1.05$ | $48.43 \pm 0.99$ | $33.79 \pm 4.35$ | $38.89 \pm 0.59$ |
| 0.400 | $43.13 \pm 0.96$ | $42.87 \pm 1.68$ | $29.71 \pm 4.56$ | $34.27 \pm 0.17$ |
| 0.500 | $38.03 \pm 1.30$ | $36.97 \pm 1.80$ | $26.06 \pm 3.60$ | $29.49 \pm 0.31$ |
| 1.000 | $16.06 \pm 1.20$ | $15.96 \pm 1.56$ | $13.61 \pm 1.22$ | $14.24 \pm 0.68$ |
| 1.200 | $11.94 \pm 0.93$ | $11.88 \pm 0.53$ | $11.20 \pm 0.95$ | $11.53 \pm 0.59$ |
| 1.400 | $10.02 \pm 0.69$ | $9.93 \pm 0.34$ | $9.81 \pm 0.82$ | $9.90 \pm 0.83$ |
| 1.500 | $9.76 \pm 0.61$ | $9.75 \pm 0.22$ | $9.46 \pm 0.77$ | $9.67 \pm 0.77$ |

