# OpenReview forum: "Benefits of Max Pooling in Neural Networks: Theoretical and Experimental Evidence"
_TMLR — Accepted by TMLR_

### Review · Reviewer_ujds · 2023-07-02

**Summary Of Contributions:**

The paper aims to analyze how valuable max-pooling is for a network architecture and asks how well max-pooling can be approximated by a ReLU network. The work introduces a function class of subpool max averages $M_d(R)$ and uses it to prove that the error when approximating a maximum of $d$ inputs using linear combinations of maximums of $ < d$ inputs is $O(1 / 2^d)$. Based on this, it is concluded that there are cases when max-pooling is strictly more expressive than a ReLU approximation. Further, the paper presents experiments confirming that increasing the width of a ReLU network does not reduce the approximation error.

**Audience:**

Yes

**Claims And Evidence:**

Yes

**Requested Changes:**

**Crucial**

C1. Add the discussion around maxout networks.

I suggest adding information about maxout networks to the Related work section to fit the results into the previous literature properly. I propose the following works, starting with [1], which introduces maxout units. Then expressivity of maxout networks in terms of the number of linear regions of maxout networks was considered in [2, 3, 4], with tight bounds being derived in [6]. Further, similarly to work by Hanin \& Rolnick (2019) for ReLUs, which is mentioned in the Related works section of the paper, the expected complexity of maxout networks was studied in [5, 7], with [7] mentioning max-pooling but not providing results specific to it.

C2. Add a discussion of how deep and wide a ReLU network has to be to attain the errors in Theorem 4 or a clear statement that it is impossible with the suggested framework.

**Minor**

Below I outline several small issues in the presentation that I found somewhat confusing while reading and that would be nice to address.

C3. More explanation around the meaning of RELERR and why considering it is meaningful would be helpful. The paper states, "This measure isolates some of the difficulties that a standard stochastic gradient descent (SGD) procedure might have in modelling max pooling operations." This statement is broad, and it needs to be clarified what kind of difficulties it refers to. Expanding on it and maybe providing examples of what kind of difficulties can be isolated this way would aid understanding.

C4. I have found the term "purely feedforward ReLU networks" somewhat unusual. As far as I understand, the paper uses networks that are usually referred to as "feedforward fully-connected networks." If my interpretation is correct, I suggest omitting the confusing term and stating equation (11) earlier in the paper, where currently, "purely feedforward ReLU networks" are defined.

C5. The name "Features" for one of the columns in Table 1 was confusing since it is unclear what is meant by stating that $M_d(R)$ is a feature. I suggest renaming it.

C6. Figure 4 would benefit from clearly writing out what each axis represents.

C7. I noticed several typos. Specifically,

- Second paragraph from the bottom of page 4: "the J1-subset max but simpler ..." --> "J1-subset max is simpler ...".

- Last paragraph on page 4. $\binom{r}{d}$ --> $\binom{d}{r}$.

- The last sentence in footnote 2 seems to be unfinished. It is unclear otherwise, on what does $G_d(R, \mu)$ impose no low-rank structure.

[1] Goodfellow, Ian, et al. "Maxout networks." International conference on machine learning. PMLR, 2013.

[2] Pascanu, Razvan, Guido Montufar, and Yoshua Bengio. "On the number of response regions of deep feed forward networks with piece-wise linear activations." arXiv preprint arXiv:1312.6098 (2013).

[3] Montufar, Guido F., et al. "On the number of linear regions of deep neural networks." Advances in neural information processing systems 27 (2014).

[4] Serra, Thiago, Christian Tjandraatmadja, and Srikumar Ramalingam. "Bounding and counting linear regions of deep neural networks." International Conference on Machine Learning. PMLR, 2018.

[5] Tseran, Hanna, and Guido F. Montufar. "On the expected complexity of maxout networks." Advances in Neural Information Processing Systems 34 (2021): 28995-29008.

[6] Montúfar, Guido, Yue Ren, and Leon Zhang. "Sharp bounds for the number of regions of maxout networks and vertices of Minkowski sums." SIAM Journal on Applied Algebra and Geometry 6.4 (2022): 618-649.

[7] Tseran, Hanna, and Guido Montúfar. "Expected Gradients of Maxout Networks and Consequences to Parameter Initialization." arXiv preprint arXiv:2301.06956 (2023).

**Strengths And Weaknesses:**

**Strengths**

S1. The paper theoretically analyses max-pooling, which has been receiving limited attention. The work also brings up the valuable point that max-pooling is often omitted from modern CNNs as an additional motivation for studying how important max-pooling is.

S2. The paper obtains new precise error bounds for the approximation of max-pooling with the composition of ReLUs and linear operations, which have not appeared before in the most closely related recent work by Hertrich et al. (2021).

S3. Generally, the theoretical part of the paper was fairly easy to follow. I have looked through the derivations, and they appear to be correct. (I have not checked the details in Appendix too closely.) I also appreciated the proof sketches and explanations of several expressions in words.

**Weaknesses**

W1. The Related work section needs to mention previous works on the maxout units for completeness.

Max-pooling is closely related to maxout units [1], which also take the maximum of several inputs. Hence, results on the maxout are closely related to the max-pooling analysis, and while max-pooling has been investigated very little, maxout has received slightly more attention. I make more detailed suggestions below in the Requested changes in item C1.

[1] Goodfellow, Ian, et al. "Maxout networks." International conference on machine learning. PMLR, 2013.

W2. The paper can benefit greatly from an explanation or an intuition of how deep and wide a ReLU network has to be to obtain approximation errors specified in Theorem 4.

The effects of the network width and depth are usually of interest in the works on network expressivity, including, for instance, Serra et al. (2018), Arora et al. (2018), Bartlett et al. (2019), and Hanin \& Rolnick (2019) mentioned in the paper, and would make the comparison of ReLU and max-pooling expressivity more fine-grained. It is not entirely clear to me if this is possible with the results in the paper, but then a clear statement about it, for instance, in the Conclusion, would add to the clarity. I could not find a statement like this in the manuscript, but if it is already present somewhere, then making it more prominent would be valuable.

**Conclusion**

Overall, the paper derives new, precise error bounds for the approximation of max-pooling with ReLU networks. The presentation is mostly clear, and the experiments, though simple, support the conclusions of the article. I believe understanding the exact effect of elements of the network architecture, such as max-pooling, is a valuable question to study. My main concerns are the incomplete review of related work and the absence of a statement quantifying in some way how deep and wide a ReLU network has to be to attain the errors appearing in the results.

---

> ### Author Response · Authors · 2023-07-24
>
> Thank your for your careful reading and thoughtful review. We are happy that you have understood the paper and appreciate the insightful feedback.
>
> As far as we can see, W1 directly implies C1 and W2 directly maps to C2, thus we for brevity, we address only the requested changes:
>
> -----
> C1. Agreed that this is crucial. The message of this paper is completely aligned with the literature you cite.
>
> In an important sense, our result is less ambitious than these papers: like Hertrich et. al. our analysis is entire concerned with a single scalar-valued function.
>
> The maximum number of linear regions illuminates our reults, though.
>
> For a single rank k max-pool layer, single output network with input dimension k, the maximal number of regions is, from [4] Theorem 8:
>
> $1 + (k - 1) k / 2$.
>
> Whereas Montúfar (2017)'s bound (since the bottlenecking correction from [4] makes this back-of-the-envelope calculation more complicated without much additional insight we prefer this looser bound) for a single-output, k-input fully connected network of width n_l at the lth layer, l = 1, 2, ..., L is
>
> $(1 + n_1) ... (1 + n_L)$.
>
> Comparing these two terms gives us a rough indication of when and how width contributes to expressivity in approximating the max function. This is nice because it gives some needed intuition on the importance of width for the experiments in Section 5.2. We will also add a short note about the [4] complexity bounds for the various ReLU network approximations.
>
> The expected number of linear regions is perhaps more important, since it is more directly applicable to practice than upper bounds arising from hyperplane arrangements. It is interesting that although the maximal number of linear regions for simple maxout networks is relatively high, the expectation is more modest, <= k in our situation [5]. This is consistent with our observation that max pooling can be difficult to approximate, but still empirically not necessary for good model performance.
>
> Finally, and this may be the most useful aspect of our work to the results on linear regions, it also suggests a whole new class of results: those that eschew general maxout assumptions for more restrictive architectures consistent with max-pooling (e.g. the output is stricly smaller than the input, inputs are grouped into disjont "pools", etc.), as although more general and canonical, maxout never seemed to gain much traction whilst max-pooling preceded it by decades and is still relatively more used (maxout is not implemented natively in PyTorch, e.g.).
>
> Thank you also for making us aware of [6] and [7].
>
> -----
>
> C2. You are right that further analysis of the necessary widths would be very interesting. Figure 4 was our attempt to collapse all possible models into parameter count, but really it only scratches the surface.
>
> Depth is pretty easy, it always requires floor(log2(max(R) - 1)) + 1 ReLU layers to evaluate $f_R^\star$.
>
> Algorithm 1 in the appendix efficiently (though not necessarily optimally) constructs a the network for a given estimator. It can be directly applied to compute widths.
>
> We propose to add plots of the widths and depths needed to implement the estimators described in Theorem 4, namely with $R = \\{0, 1, d - 1\\}, \\{0, 1, d - 2, d - 1\\}$, and $\\{0, 1, 2, ... , d - 2, d - 1\\}$. This would augment Section 5.3 and also help to better explain the x axis for the plots in section 5.2.
>
> About the last: as can be seen, the network size grows very quickly with $d$, and the intuition is simple: we have to compute a significant fraction of the ${d \choose j}$ subpool maxes at the $j$th layers, thus in total, we have to compute essentially $\sum_{j=1}^d {d \choose j}= 2^d - 1$ terms.
>
> This is what we meant in the abstract by "[the] residual can be made exponentially small in the kernel size, but only with an exponentially wide approximation". Unfortunately, this point was inadequately developed in the paper previously. We will correct this.
>
> For the plot, we thus restrict $d$ to a value that can be evaluated quickly on a laptop. In theory, it should be possible to compute analytically the layer sizes in a way that avoids their actual construction, but it was not straightforward after a few hours of analysis. If you feel like this would be a valuable addition to the paper, we are confident that it can be added, though it may be inelegant.
>
> C3 - C7:
> We uniformly agree and have actioned them.
>
> Especially C3, the point there being that accurate approximations of the maximum entail many offsetting contributions from different order statistics. This entails relatively high norm weights, and also many directions in both input and weight space along which the estimator does not change. In turn, this suggests a high sensitivity to small perturbations during training and initialization. For this, we should draw some lessons on the importance of these factors to the number of activation regions, for example [7].

---

> > ### Comment · Reviewer_ujds · 2023-07-27
> > **Reply to the Comment by Authors**
> >
> > Thank you for the detailed reply addressing all the points from the review. I also appreciated the new plots and the additional note explaining how the results in the paper would be more specialized than the maxout results.
> >
> > The proposed changes will address the issues raised in the review, and I have no other concerns.

---

### Review · Reviewer_W2yX · 2023-07-30

**Summary Of Contributions:**

The paper provides a detailed theoretical analysis of the characteristics of max pooling in deep neural networks and investigates possible approximations of it, such as using compositions of relu activations. They introduce a more general class of approximations, subpool max averages, which take the max over subsets of the features and average them together. They provide analysis that the approximation they propose, cannot fully approximate the max pooling operation, however, it can get exponentially close. They provide some experimental evidence to find their claims, mainly that the max pool operation cannot easily be substituted by other operations.

**Audience:**

Yes

**Broader Impact Concerns:**

N/A.

**Claims And Evidence:**

No

**Requested Changes:**

Following point 1 above, I believe that the approach of providing experimental evidence should be reversed, rather than showing that there is no statistical significance to rule out the hypothesis (that increasing mu could improve error rates), it will be better to show that there is statistical significance of the opposite claim.  I am not sure about what setup should be taken here, but I believe that the results should show some statistical significance. Either way, I do not think that the current experimental evidence is sufficient in this manner, please provide perhaps a different experiment with statistically significant results, to support the claims of this paper.


Please provide more details on the experiment setup, the model architecture specifications, a diagram, and preferably code with data. In its current form, it is a bit difficult to understand the experimental setup. If you decide to run a different experiment as mentioned above, please add more details there.


**Strengths And Weaknesses:**

Strengths:

Well written.

Main Weaknesses:

1. It is not clear to me that the max pooling being a function that cannot be easily approximated is indicative that it is a useful operation in modeling real-world problems in machine learning. I believe that this can be tested experimentally pretty easily, by swapping out different pooling operations, as is done in [1]. I would appreciate more motivation to support this line of work.

2. It is not clear to me that the experimental evidence clearly supports the claims. Though the authors aim to show that no significant improvements in error rates are achieved by increasing the parameter mu, there can be several other contending hypotheses that explain this result, for example, failed optimization, bad architecture design, data generation problems, etc.

[1] Williams, T. and Li, R., 2018, February. Wavelet pooling for convolutional neural networks. In International conference on learning representations.

---

> ### Author Response · Authors · 2023-07-30
>
> Thanks for your review. What is "[1]" (from " by swapping out different pooling operations, as is done in [1]"), please?

---

> > ### Comment · Reviewer_W2yX · 2023-07-30
> > **Missing citation in review**
> >
> > Sorry, copy-paste error. Now updated.

---

> ### Author Response · Authors · 2023-08-29
>
> Thanks for your review.
>
> I will just directly speak to the weaknesses, then recap any requested changes that have not been addressed yet.
>
> Weakness #1:
>
> Thank you for making aware of [1] -- we agree that it is a pertinent study. We were not able to incorporate a citation into the paper in a way that was not too stilted, but we’ve read and understood it.
>
> Happily, I think this perceived weakness arises mostly from a misunderstanding.
>
> We are not claiming -- and our paper does not require -- that max pooling is a useful operation in modelling real-world problems in machine learning. We are agnostic about its actual utility. All we require is that in an earlier generation of computer vision models, it was often used, and more recently it is not.
>
> We've clarified this in the updated version of the paper we will upload shortly.
>
> But here's the essence: AlexNet dominated the ImageNet Large Scale Visual Recognition Challenge (ILSVRC) 2012, and had max pooling in more than half of its "blocks". The leaderboard is here: https://image-net.org/challenges/LSVRC/2012/results.html (AlexNet is a product of the "SuperVision" team). However, the ILSVRC 2017 leaderboard was dominated by Resnet-variants, for which max pooling is not central. Similarly with DawnBench (https://dawn.cs.stanford.edu/benchmark/#imagenet), where a more transparent source code distribution model makes it simple to verify that many models feature only a single max pooling layer.
>
> A statement corroborating our assertion that max pooling used to be often used is from the first sentence of Springenberg et al. (2015):
>
> "The vast majority of modern convolutional neural networks (CNNs) used for object recognition are
> built using the same principles: They use alternating convolution and **max-pooling** layers followed
> by a small number of fully connected layers..." [emphasis ours]
>
> Hopefully this comprises the missing motivation? We do not believe it necessary to establish that max pooling outperforms other pooling methods, akin to [1]. Indeed, by highlighting the fact that max pooling has seen declining use, we are implicitly (and explicitly, in the abstract: "Since max pooling does not seem necessary, "), saying that max pooling is _not_ an indispensable operation in modeling real world problems. So really, as we understand the exercise you proposed, we do not expect to see what you think would support our argument.

---

> > ### Author Response · Authors · 2023-08-29
> >
> > Weakness #2
> >
> > You are correct: it may be possible to decrease error with more capacity, and our experiment fails to achieve it.
> >
> > We could not develop an experiment that could directly demonstrate the desired conclusion (and would not be surprised if it were impossible!). It is true that our result is awkward because failing to achieve low error on a deep learning task is not difficult, for example it could result from a coding mistake or a bad hyperparameterization.
> >
> > However, there is an important precedent for this type of result in machine learning experiments: whenever a paper shows that some method is bested by another, we must assume that the inferior method implemented correctly and reasonably. Thus, although demonstrating experimentally the correctness of a failing method requires a high standard of evidence, the nature of the argument is not inherently problematic. Basically, advancing science requires believing these sorts of results, though it is reasonable to hold them to a higher standard of evidence. And best practice is sensible: model the problem simply, implement the experiment in an idiomatic fashion, check that the model can solve problems that should be solvable, and transparently release the source code. We have done all of these things.
> >
> > To your specific concerns (in decreasing order of significance):
> >
> > (1) "bad architecture design": Cannot be an alternative explanation of our results. This is by construction -- since the theoretical results only concern fully connected networks, we only need to experimentally analyze these architectures.
> > For sure, it is possible to relax $\mathcal{M}(R)$ other than with by a proportional increase in width. However, depth
> > is fixed, and thus at some width multiplier, the scheme adopted in our experiment will be more expressive than any other
> > scheme (because fully connected networks are completely characterized by their layer dimensions). We will make this
> > point in the paper, since it is important.
> >
> > (2) "data generation problems": Because our theory is developed only for data on the unit cube, there is limited scope
> > for exotic data problems. Nonetheless, there may be some subtle bias so it is worth a sanity check. Thus, we also examine
> > the Dirichlet distribution (which emphasizes more mass around the boundary) and also a Sobol point set as a more uniformly
> > distributed set of data. These experiments are described in the paper.
> >
> > (3) "failed optimization" is, in our opinion, the key point, though we construe it more broadly as "failed fitting".
> > Our approach is sound, but for sure it is responsibility to comprehensively convince you and the audience that our
> > results are robust to different modelling choices and reliably represent what would be seen in a general application. For this, in the paper we will upload shortly, we add to the appendix a fairly comprehensive set of robustness checks around initialization, optimizer, etc.
> >
> > I don't know if the point on statistical significance is pertinent in light of our above clarification, but we should mention that the low uncertainty quantified in our results suggest a high level of significance.
> >
> > Finally, what sort of diagram do you envision? The code (including the code that generates the data) was already submitted in the supplementary materials. See the "readme.txt" there for more details.

---

> > > ### Author Response · Authors · 2023-09-01
> > >
> > > Hi, sorry for the spam. We have uploaded an updated version of the paper as the most effective way to update the empirical results. A high level description of the changes and more on the motivation is described here: https://openreview.net/forum?id=YgeXqrH7gA&noteId=y276K4Pcu4. Comments specifically and exclusively addressing our discussion are in red, and probably several of the points are discussed in yellow also (meaning that the text addresses the feedback of >= 2 reviewers).
> > >
> > > Thanks!

---

### Review · Reviewer_SozV · 2023-08-10

**Summary Of Contributions:**

This work focuses on the question of how well the max function can be approximated by feedforward ReLu networks. The work claims to draw inspiration from the fact that max pooling used to be prevalent in multilayered CNNs on image tasks, but have now given way to only a single pooling layer at the end.

The authors first go through a series of basic constructions showing that as the number of elements to be maxed over increases, constructing a max function with feedforward ReLus requires deeper networks (both for efficiency but also to correctly implement the functions). They then perform an analysis about how well the true max is approximated with subpool maxes - maxes over subsets of the data. They characterize the error generated from subpools of different sizes, and show that the worst case error is achieved over a non-trivial volume of space with inputs on the hypercube.

Finally, the authors study numerically how SGD solutions compare to approximations from subpool maxes. They show that even at widths larger than those required by optimal subpool max approximations (constructed using an algorithm derived by the authors), ReLu networks trained with SGD don't improve their approximation with width and remain far from the true max function.

**Audience:**

Yes

**Claims And Evidence:**

Yes

**Requested Changes:**

In section 4, Theorem 5 seems to be more interesting/relevant to me than theorem 4, especially given that some of the results of theorem 4 (particularly, equations 5, 6, and 10) can be derived fairly simply even without the definitions and framework presented by the authors. The proof of theorem 5 is buried in the appendices and follows from a long string of other theorems. I think that it is important to do the following:
* Expand the discussion of the theorem, and why it holds. This will bring both greater clarity to the question of the "typical" approximation error, and is a much more relevant to the readership of TMLR.
* Better signpost/lay out the groundwork for proving theorem 5 in the appendix, before going through all the related theorems. As it currently reads, theorem 5 appears very quickly from the rest of the theorems, which seems nice but makes it hard to read those appendices and to understand _why_ the results are true. The why is important in this case because readers may wish to use similar techniques to study other function approximation problems.

I did not understand the argument in Section 5/Appendix E that the algorithm is efficient. It was claimed that the algorithm runs in $\binom{d}{r}$ time which, for $r=d/2$, is exponential in $d$ This seems very slow. Some more detailed discussion on this point would be greatly helpful. The algorithm should be described in more detail in the main text, as it builds on some of the previous results.

Some discussion about how max is/could be implemented with other non-linearities, particularly doubly saturated ones like tanh, and attention like blocks would be greatly helpful in positioning this work appropriately in the literature.

Please make sure that the final version has proper hyperlinking to the Appendix; I recognize that this version did not as the main text and appendix were in separate files.

All of the plots need properly labelled axes so that readers can get some understanding semi-independently from the captions.

For the analysis in Section 5, some discussion/experiments on whether or not the networks are nearing the NTK/kernel regime for larger mu values would be helpful for the discussion.

I believe that "confidence interval" is the wrong way to describe the variability Figures 2 and 3, as it is true variability in a learning procedure (SGD). Indeed, over 10 almost certainly non-Gaussian datapoints, the relationship between sample standard deviation and the cdf does not hold (aka z-scores are not the correct framework here). I would recommend plotting the standard deviation itself, or the min and max values over the 10 seeds.

**Strengths And Weaknesses:**

The main strengths of the paper are the relatively simple and clean definitions and constructions. In particular I find the subpool analysis quite interesting, and it makes me wonder if there are other problems in deep learning theory which could be attacked by looking at statistics over subsets of neural activations.

The main weakness of the work is the relevance to the audience of TMLR. 'm not sure of the overall interest in characterizing the ability of ReLu networks to implement the max function on high-dimensional inputs. The work is self contained and thorough and I think would be valuable to anyone thinking about these issues. Some of the frameworks may be more broadly useful to the theory community.

However, the results of section 5 seemed more confusing and of les relevance. One thing that was very unclear to me was whether or not the SGD trained models were approaching the wide network/NTK regime. In that case, it would be completely unsurprising that as width goes, the approximation performance remains roughly constant; indeed, this has been observed at widths as small as ~ 500 in certain problems. It is definitely impossible to learn the max function in that regime, at least without resorting to exponentially many samples. It also is not very clear to me what to think about plots for a fixed, small number of maxed elements d - in practice one might care about larger d, or what happens as d increases.

More generally, the max function is usually a "means to an end"; namely getting good performance in, say, a supervised learning setting. In this case, it is not clear what approximation metric is actually relevant. It could be that the max doesn't need to be approximated well (or indeed, may not be needed at all!). This in my mind reduces the motivation/relevance, as in the case of max-pooling the max was chosen more as a useful tool rather than the end and of itself.

Another issue in section 5 is that the "confidence intervals" go almost all the way to 0 in some cases - in other words, the variability of the losses is as large as their typical values. It's possible that with some sort of hyperparameter tuning or cross validation/early stopping, the losses could be dramatically lowered in those cases; these types of procedures are common in normal machine learning and may better characterize the relationship between the exact solutions and the ones found by SGD.

For now I have marked "no" for relevance to the audience of TMLR, but I am willing to change my mind on this. I look forward to a productive discussion with the authors and other reviewers on this point.

In addition, I have marked "no" for claims and evidence due to it being somewhat hard to parse some of the results. I have some requested changes which I believe are tractable and should flip that to a "yes".

Update: after reading the other reviews and discussion with the authors, I have flipped both relevance and claims and evidence to yes. However I still strongly suggest that the authors give more intuition for the proof of theorem 5 in the main text.

---

> ### Author Response · Authors · 2023-08-19
> **confusion on the NTK point**
>
> Thanks for your very detailed and thorough review.
>
> In order to give you the best response, please can we clarify one point at this early stage?
>
> You say:
>
> "One thing that was very unclear to me was whether or not the SGD trained models were approaching the wide network/NTK regime. In that case, it would be completely unsurprising that as width goes, the approximation performance remains roughly constant; indeed, this has been observed at widths as small as ~ 500 in certain problems. It is definitely impossible to learn the max function in that regime, at least without resorting to exponentially many samples."
>
> Our understanding of the NTK result is that for very wide (at a given dataset size, that is, "overparameterized") networks, the global optimum will be achieved by SGD? This would seem to suggest that the error _should_ fall? Are we mistaken in our understanding about what the NTK means? Or are you drawing some different implication? We note as well that your statement above seems to directly directly contradicts Reviewer Sqmz?
>
> You might find useful the plots of layer widths that we prepared for Reviewer ujds (in the supplementary materials)?
>
> We will definitely look at varying the number of samples in our experiments -- this had not crossed our mind. If there is a simple test that can be done to help clarify your thinking please mention it!

---

> > ### Comment · Reviewer_SozV · 2023-08-20
> > **NTK point**
> >
> > The part of the NTK regime i'm referring to is: when networks start becoming wide, they are well approximated by a linearized model, where the Gram matrix is given by the NTK at initialization. Therefore, as width increases, you are essentially training a better and better approximation to some limiting linear model.
> >
> > This means that at some point there is no benefit to increasing width while leaving e.g. the dataset intact. On a dataset like MNIST, some of this effect can already be seen as network width approaches, say, ~ 1000 or so. This to me might provide an explanation for why at some point increasing width doesn't change any of the metrics much.

---

> > > ### Comment · Reviewer_SozV · 2023-08-20
> > > **Intuition about NTK point**
> > >
> > > I see now the other comment suggesting that in the NTK limit one might actually expect the error to drop to 0. This is true of a training error, but not of a generalization error. Also, this is only true if the NTK has full rank; if the NTK has any 0 eigenvalues, then the training error will go to a constant. Is the error plotted in the figures equivalent to training error or test error?
> > >
> > > One measurement which could clarify this point would be to simply measure the NTK during training, and see if the correlation between the final NTK and the initial NTK is high.

---

> ### Author Response · Authors · 2023-08-29
>
> Thanks for clarifying about the NTK. We've managed to compute the NTK of the model as outlined here https://pytorch.org/functorch/stable/notebooks/neural_tangent_kernels.html, and there does not appear to be any zero eigenvalues. We're a little unclear about the correct way to measure the correlation between NTKs, but before diving into it further we thought it would be worthwhile presenting our other work on the NTK, because we do have some insights.
>
> Rather than promise to better signpost or whatever, we will in the next few hours upload an updated version of the paper that directly implements the requested changes. To your points: we have better motivated the main theorem nearer to its statement, and added a description of the argument from the appendix in words to better lay out the argument so that the idea can be had without the need to work through the appendix. It's great that you find the method useful, but we understand that not everyone will, and in the revised version it should only be necessary to look at the appendix if someone wants the full rigor.
>
> We've also attempted to better introduce the result immediately before it appears. Please let us know if this is not adequate in your opinion.
>
> Your point about efficiency is correct and well-taken. Basically, given the need to compute ${d \choose r}$ terms it is efficient in that through careful it can skip computations that would be performed in a naïve implementation, and it can reuse earlier calculations in a (we think) fairly elegant manner. But yes, fundamentally, it is exponentially complex in $d$ as you point out. This has been clarified and a bit more discussion of the algorithm has introduced to the main text.
>
> We've added a bit of discussion about the extension of the result to different activations. Basically, we expect that except for attention, our results all go through. We don't have much to say about attention, unfortunately, though we do observe that max pooling is absent from the Torchvision implementation of the Vision Transformer.
>
> The hyperlinking will work. Basically it was broken in the submission because we wanted it to work in the final paper: we compiled the paper and appendix as a single file, then use pdftk to split it.
>
> The updated plots are much more descriptive.
>
> Agreed that 10 PRNG seeds is too few to invoke the Central Limit Theorem. We've eliminated this language and added the min and max as your suggest. Interestingly, the min and max appear to largely coincide with the mean $\pm$ 1.96 standard deviations so (since we'd expect these to be >3 standard deviations away) the "confidence interval" interpretation does actually seem to be supported. That said, it's a small matter, though rather than $\pm $ 1 standard deviation, we elect to stick with 1.96, so that a reader who wishes to interpret the shaded area as a confidence interval can. This clearly just differs from your suggestion by a constant multiple (for better or worse, 68% e.g. the confidence associated with $\pm 1$ standard deviation of a Gaussian is not a very interpretable quantity).
>
>
> So: please have a look at the version of the paper we will be uploading later today, and please can you elaborate on the correct way to compute the correlation between the initial and terminal NTKs if your concern was not addressed by our other results?

---

> > ### Author Response · Authors · 2023-09-01
> >
> > Hi, sorry for the spam. We have uploaded an updated version of the paper as the most effective way to update the empirical results. A high level description of the changes and more on the motivation is described here: https://openreview.net/forum?id=YgeXqrH7gA&noteId=y276K4Pcu4. Comments specifically and exclusively addressing our discussion are in green, and probably several of the points are discussed in yellow also (meaning that the text addresses the feedback of >= 2 reviewers).
> >
> > Thanks!

---

> > > ### Comment · Reviewer_SozV · 2023-09-05
> > > **Comments on new draft**
> > >
> > > Thanks for the new draft; the color coding in particular is quite helpful!
> > >
> > > Comments:
> > > * Citations should include: https://proceedings.neurips.cc/paper_files/paper/2019/hash/0d1a9651497a38d8b1c3871c84528bd4-Abstract.html
> > > * At infinite width, as long as the kernel isn't degenerate, training error is always 0. That doesn't seem to be the case in your setup from Figure 4, suggesting the experiments are NOT in the NTK regime, unless there was some issue with the optimization choices/regularization.
> > > * Figure 4 is much improved; I would just shade from the min to max region rather than also include some confidence interval type shading (since variance estimates over 10 samples will be incredibly noisy to begin with).
> > > * I still think Theorem 5 needs better explanation in the main text.

---

> > > > ### Author Response · Authors · 2023-09-06
> > > >
> > > > Yes, that’s a very nice paper, with some very clear and elegant intuition. Thank you for suggesting it. We will add it.
> > > >
> > > > It seems like, indeed, we are not in the NTK regime. And I think this is okay and not unexpected. Foremost, this is because we are only trying to observe empirically that error cannot be greatly lessened with a small increase in capacity, which we have done. It’s not unexpected because the networks we are approximating with are not that wide, even relative to those discussed in Lee 2019 (e.g. widths of between 2048 and 8192), for example with $\mu = 4$, the $R = \\{0, 1, …, d - 1\\}$ approximation has widths [456, 2032, 1504] for the experiments in $d = 8$ shown in Section 5.2.  For what it is worth, we can get the train error to essentially zero for very small dataset sizes (e.g. 100 rows). We interpret this as some evidence that the optimization generally works, and that impractically large models are needed to get into the NTK regime.
> > > > A slight caveat to this last point is that we are also not doing the NTK parameterization (which should not be necessary from what I can tell).
> > > >
> > > > In response to this question, we also ran some experiments to see whether much smaller learning rates could better put us into the NTK regime – it could not. Even lowering the learning rate by a factor of 10 did not substantively change the $\mu$ vs error plots. We will add these results to Appendix F.
> > > >
> > > >
> > > > On the uncertainty bands: we will drop the $\pm$ 1.96 standard deviations.
> > > >
> > > > We added a bit more discussion about Theorem 5 to draw an analogy to the measure-zero sets seen in the literature on piecewise linear regions. It’s honestly a bit of a technical afterthought, and with much further discussion it may not be justifying the space. What would you think to add? Alternatively, maybe it should be moved to the appendix?

---

> > > > > ### Comment · Reviewer_SozV · 2023-09-12
> > > > > **Discussion of theorem 5**
> > > > >
> > > > > I guess i was looking for more intuition on why theorem 5 is true, not justifying the calculation. I do think it's quite an important result, as much ML theory suffers from being true on a set of measure (close to 0) and does not well characterize typical behavior. In some sense, theorem 5 is what convinced me personally that the theoretical analysis as a whole was useful. I still think it should be emphasized more/the proof idea explained a bit better in the main text.

---

> > > > > > ### Author Response · Authors · 2023-09-13
> > > > > >
> > > > > > Right, makes sense. Thanks for your faith in that result :).
> > > > > >
> > > > > > I think it'll be good to add a plot of the L1 distance between a few different estimators (for different $R$) and the actual max along with the description of the experiment it should give a top-to-bottom elaboration of the idea of the proof. I started working on it today but had something urgent come up -- will finish and upload the fix tomorrow.

---

> > > > > > > ### Author Response · Authors · 2023-09-14
> > > > > > >
> > > > > > > I did the experiment and found something quite cool. Actually the empirical $L_1$ error seems to be basically uniformly distributed between zero and the $L_\infty$ error treated in the paper. You can see this in the plots I added to the supplementary materials, where I generate 10000 Dirichlet(1,...,1) distributed random variates and just look at the empirical error for three different $R$. So, for example, almost surely the error is nonzero, and the average error seems to actually be the max error / 2.
> > > > > > >
> > > > > > > It is reasonable in retrospect but I had not intuited it beforehand. Do you think a discussion of this would sufficiently emphasize the point? We could also attempt a formal proof of this observation if you think it'd be interesting.

---

> > > > > > > > ### Comment · Reviewer_SozV · 2023-09-14
> > > > > > > > **Thanks for the additional experiments**
> > > > > > > >
> > > > > > > > I think this is quite nice! Please add it to the discussion. I wouldn't say know to a proof but I know the devil's in the details - maybe make a good faith attempt, and if not add discussion of the numerical results to the main text?

---

> > > > > > > > > ### Author Response · Authors · 2023-09-18
> > > > > > > > >
> > > > > > > > > Just a quick update in case a decision is incoming. tl;dr is that I have the proof I wanted.
> > > > > > > > >
> > > > > > > > > As I said above, the uniform error condition is under a Dirichlet distribution of data over the unit cube. The intuition is that Dirichlet distribution has more mass in the corners, which is where the error is higher. The uniform distribution might actually be a sort of optimality condition, but I've not thought too much about it.
> > > > > > > > >
> > > > > > > > > This is because, after a bit more thought, I think that it is somewhat preferable to analyze the error for uniformly distributed data. Both are fine for making the point about the error being almost surely nonzero, so this is purely a rhetorical choice -- basically if any will do I'll use the simpler and better understood distribution. For a uniform distribution, the error distribution is more-thin tailed, but still symmetric, etc. I've added the analogue of these plots to the supplementary materials.
> > > > > > > > >
> > > > > > > > > Happily, I learned that the distribution of positively-weighted order statistics is known, cf. [1]. The distribution is relatively simple (once one groks all the notation) and fully closed-form -- say 10 lines of Python/numpy. The error of our estimator is a general (not positively-weighted) linear combination of order statistics, however there is a reduction to the positive-weighted core result that relies on exchangeability from [2]. The argument is unfortunately a little bit complicated, but also simple enough. I have coded both and can analytically compute the CDF of a general linear combination of order statistics. I have extensively tested that these formulae agree with simulations and am totally confident that I have worked through the somewhat confusing notation and such.
> > > > > > > > >
> > > > > > > > > I will work on incorporating this over the next few days, but to minimize noise I'll not update the paper here or reply again unless you want to see it before making a decision :).
> > > > > > > > >
> > > > > > > > >
> > > > > > > > > [1] Weisberg, Herbert. “The Distribution of Linear Combinations of Order Statistics from the Uniform Distribution.” The Annals of Mathematical Statistics 42, no. 2 (1971): 704–9. http://www.jstor.org/stable/2239815.
> > > > > > > > >
> > > > > > > > > [2] Morganna Carmem Diniz, Edmundo de Souza e Silva, H. Richard Gail, (2002) Calculating the Distribution of a Linear
> > > > > > > > > Combination of Uniform Order Statistics. INFORMS Journal on Computing 14(2):124-131. https://doi.org/10.1287/
> > > > > > > > > ijoc.14.2.124.121

---

### Review · Reviewer_Sqmz · 2023-08-13

**Summary Of Contributions:**

This paper studies the question: 1, Can max pooling be replaced by linear mappings with ReLU activations? 2, When would doing so give a different model? This paper introduces the subpool max functions to show the max function cannot be approximated by the linear combinations of subpool max functions. The analysis is also extended to wider neural networks by experiments.

**Audience:**

Yes

**Claims And Evidence:**

Yes

**Requested Changes:**

1. I suggest that this paper can be improved by implementing experiments on real-world datasets. For example, showing the results of Figures 2 and 3 using MNIST and CIFAR-10 would be better.

2. The experiments in Section 5.2 shows the error does not change much as the networks become wider. This is different from the conclusions from NTK. Some clarifications and discussions are needed here.

3. From my understanding, the max-pooling function is still widely used in CNN-based neural networks. I feel it is not accurate to say modern networks have few max pooling functions. Also, supporting examples like ResNet, MobileNetV3 were at least four years ago, and I don't think they are that modern. I am wondering whether Transformers have this proposed trend of the max-pooling function. Some discussion is needed.

**Strengths And Weaknesses:**

Strengths:
1. The logic of this paper is clear, and the conclusions seem correct and reasonable.
2. The topic is interesting to me.

Weaknesses:
1. The motivation is a little strange to me.
2. The experiments are not so enough.

---

> ### Author Response · Authors · 2023-08-19
> **confusion on the NTK point**
>
> Thank you for your review.
>
> We wonder if you can help us reason out an apparent confusion? You seem to say that error not dropping with more width is surprising because of NTK intuition, and Reviewer SovZ seems to say that it is not surprising because of NTK intuition. Do you disagree with their conclusion, or are we missing something?
>
> We are more in your camp -- we were a bit surprised just how constant the performance was with more model capacity. We expected (and we only needed for our empirical results to support our theoretical work) the error to decline at first and _eventually_ level off. We did not expect it to be constant even at networks that were only a little bit larger. In that sense, our experiments are a bit "too strong".
>
> We are running and writing up some more experiments to more comprehensively show that our result, whilst surprising, is robust. In particular, we are running some experiments on networks that are a single layer, but very wide, as this seems to be the architecture most NTK-like.
>
> I wonder, does the NTK intuition really offer anything stronger than the universal approximation theorems (described in the paper)?

---

> > ### Comment · Reviewer_Sqmz · 2023-08-20
> > **Explanation about NTK**
> >
> > I am unsure about the answer to this question, but I can show you my understanding. With the NTK theory, the test error generally decreases as the network width increases, but at a sublinear rate. Please see Figure 1(a) of [1] and Figure 2 of [2]. These are two theoretical works on NTK. The error always decreases but does not change much when the network width is large enough. I suggest you cite these two works as references for a clear explanation of this discussion.
> >
> > I am confused about your last question. NTK is more about convergence and generalization. It is more about the learning process. However, from my understanding, the universal approximation theorem you use is about the approximation error/expressive power of the network. It is not related to the learning process. They are two different problems. Please let me know if my understanding is incorrect.
> >
> > [1] Allen-Zhu et al., "Learning and Generalization in Overparameterized Neural Networks, Going Beyond Two Layers" Neurips 2019
> >
> > [2] Li et al., "Generalization Guarantee of Training Graph Convolutional Networks with Graph Topology Sampling" ICML 2022

---

> ### Author Response · Authors · 2023-08-29
>
> Note: It does not appear possible to post detailed plots on openreview. We have thus elected to simply post a revised version of the paper that has a considerably considerably revised version of the paper to address the critiques of the experiments. Please see this.
>
> We are sympathetic to the point about the point about the motivation. Other reviewers have also highlighted how mimicing a single max operation is not exactly the point (e.g. we should probably care more about the overall performance of the network). This is entirely correct, but, put simply, other papers have already done this (see our discussion with ujds), and our approach does ultimately address the motivating question in a straightforward way that allows us to obtain fairly strong and precise results. We suggest you view our work as in the same vein as "Matus Telgarsky. Benefits of depth in neural networks." in that it sets up a too-simplistic to be off-the-shelf usable model of a ubiquitous phenomenon and ultimately reaches a nice takeaway.
>
> We have added considerable experiments to the updated version of the paper both more finely examining the NTK point and more generally presenting the result in a more direct way. But to the point about real world datasets: There is no real world corollary of our results, sadly. It would obviously be simple to set up a horse race between max pooling and linear-ReLU blocks in a way that would make for a more apparently-impressive paper (indeed, we do this in the appendix for adversarial robustness, to give some early results demonstrating that max pooling tends to produce more adversarially robust models), but I think this would ultimately be posturing that does make for a clearer story.
>
> Thank you for explaining your understanding of the NTK. To this point: please see the updated paper for comprehensive experimental results, but put simply: we did not present the $\mu < 1$ regime (it is not actually necessary to make our point), and here is where the error falls massively, akin to the plots you suggest from [1] and [2].
>
> Finally, about the claim that max-pooling is less widely used more recently: Max-pooling is still widely used in the sense that many models have it, but it is less used in the sense that often a model will have a single max-pooling layer towards the end, whereas earlier generations of models might have had several max-pooling layers, e.g. one for every two convolutions. Discussion of this point has been added to the paper. And we will also visit this point in more detail in our response to Reviewer W2yX, so please see this also. But to the point about Transformers: the Torchvision implementation of the Vision Transformer also does not use max pooling (this point is discussed in the updated version of the paper).

---

> ### Author Response · Authors · 2023-08-29
>
> It seemed most fruitful to just come back with a version of the paper that incorporates all this great feedback, so that is what I will orient the discussion around.
>
> Changes (mostly additions) or other points that specifically address the comments of w2yx are in red, of ujds in blue, sqmz in purple, and sozv in green. Fundamentally new points that address points made by multiple reviewers is in yellow. New writing that is necessary to accommodate the redone organization, but does not specifically seek to make novel point is not colored, which is to say that not all changes have been specifically highlighted.
>
> We have incorporated a comparison to work on the number of linear regions at several points throughout the analysis.
>
> Clarified the timing on "early", "more recent", etc. image classifiers and oriented attention-based models to the menagerie. Unfortunately, the work does not have anything super insightful to say about transformers, but the little that it can say is completely consistent with the existing narratives, as ViTs tend not to use max pooling either.
>
> We have attempted to clarify the figures, striking a balance between clutter and ability to stand alone.
>
> Have discussed and incorporated references to the many excellent studies that were overlooked.
>
> We have completely redone Section 5, in a way that would be too colorful to follow the scheme above. So we would request that all reviewers please view the updated results and presentation. The most substantive and pointed rejoinders were colored in order to highlight them, but really more than half of the content in this section is new and reflects many overlapping ideas. The most substantive change was the removal of the “RELERR” measure – this was a nice idea, but ultimately with the addition of so many more results and analysis it does not appear to carry its own weight (though if any reviewers disagree, we can look to put it back).
>
>
>
> We would like to add one more result and perhaps do a bit of cleanup of the writing, and the changelog will reflect this, but do not plan any substantive changes.

---

### Decision · Action_Editors · 2023-09-17

**Recommendation:** Accept with minor revision

**Comment:**

The paper makes a useful contribution by analyzing max-pooling, and the results should be interesting to the deep learning community.

The paper was reviewed by four reviewers. Initially the reviewers were concerned about whether max-pooling is necessary regardless of whether it can be approximated well by ReLU networks. The reviewers also pointed out missing references such as maxout network. After extensive discussions, all four reviewers were positive. One reviewer hoped to see theorem 5 referenced more in the main text of the revision.

**Audience:**

The paper should be interesting to the deep learning community in general, especially as max-pooling is less used in modern architectures.

**Claims And Evidence:**

This paper studies the max pooling in neural networks both theoretically and experimentally, in particular, whether max pooling can be approximated well by ReLU networks. The claims made in the paper are supported by both theoretical and experimental evidences.